# Regulation of zebrafish melanocyte development by ligand-dependent BMP signaling

Alec K Gramann[1,2], Arvind M Venkatesan[1,2†], Melissa Guerin[1,2], Craig J Ceol[1,2]*

[1]Program in Molecular Medicine, University of Massachusetts Medical School, Worcester, United States; [2]Department of Molecular Cell, and Cancer Biology, University of Massachusetts Medical School, Worcester, United States

**Abstract** Preventing terminal differentiation is important in the development and progression of many cancers including melanoma. Recent identification of the BMP ligand *GDF6* as a novel melanoma oncogene showed *GDF6*-activated BMP signaling suppresses differentiation of melanoma cells. Previous studies have identified roles for *GDF6* orthologs during early embryonic and neural crest development, but have not identified direct regulation of melanocyte development by GDF6. Here, we investigate the BMP ligand *gdf6a*, a zebrafish ortholog of human *GDF6*, during the development of melanocytes from the neural crest. We establish that the loss of *gdf6a* or inhibition of BMP signaling during neural crest development disrupts normal pigment cell development, leading to an increase in the number of melanocytes and a corresponding decrease in iridophores, another neural crest-derived pigment cell type in zebrafish. This shift occurs as pigment cells arise from the neural crest and depends on *mitfa*, an ortholog of *MITF*, a key regulator of melanocyte development that is also targeted by oncogenic BMP signaling. Together, these results indicate that the oncogenic role ligand-dependent BMP signaling plays in suppressing differentiation in melanoma is a reiteration of its physiological roles during melanocyte development.

*For correspondence:
craig.ceol@umassmed.edu

Present address: †Syngene International Ltd, Bangaluru, India

Competing interests: The authors declare that no competing interests exist.

## Introduction

Tumor differentiation status is often an important prognostic factor in cancer. For many cancer types, tumors that are less differentiated are associated with a higher grade and worse prognosis compared to more differentiated tumors, which often follow indolent courses (*Hoek et al., 2006*; *Rosai and Ackerman, 1979*). In order to adopt a less differentiated state, a common event in cancer is downregulation of factors that drive differentiation of adult tissues (*Chaffer et al., 2011*; *Dravis et al., 2018*). This loss of pro-differentiation factors is often coupled with an upregulation of other factors that are associated with embryonic or progenitor states (*Caramel et al., 2013*; *Tulchinsky et al., 2014*). Thus, many de-differentiated and high-grade cancers have gene expression profiles associated with early development (*O'Brien-Ball and Biddle, 2017*).

Developmental factors and pathways co-opted by cancers are often related to vital cellular functions, such as proliferation, migration, and differentiation (*Caramel et al., 2013*; *Casas et al., 2011*; *McConnell et al., 2019*; *Perego et al., 2018*). Furthermore, the embryonic origin of specific tissues can impact the aggressive phenotypes tumors are able to acquire (*Carreira et al., 2006*; *Gupta et al., 2005*; *Hoek and Goding, 2010*). In the case of melanoma, the cell of origin, the melanocyte, is derived from the neural crest, a highly migratory population of embryonic cells. Thus, melanomas are prone to early and aggressive metastasis, associated with the expression of neural crest migratory factors (*Liu et al., 2014*). Additionally, melanomas lacking differentiation exhibit more aggressive characteristics and are broadly more resistant to therapy (*Fallahi-Sichani et al., 2017*;

*Knappe et al., 2016*; *Landsberg et al., 2012*; *Mehta et al., 2018*; *Müller et al., 2014*; *Shaffer et al., 2017*; *Zuo et al., 2018*). While differentiation status is evidently important in the course of disease, the mechanisms by which melanomas and other cancers remain less differentiated are poorly understood. Since many of the factors associated with a lack of differentiation in these cancers are expressed and apparently function during embryogenesis, elucidating the developmental roles of these factors can give insight into their behaviors and roles in tumorigenesis and progression.

A key pathway involved in early development and development of the neural crest is the bone-morphogenetic protein (BMP) pathway (reviewed in *Kishigami and Mishina, 2005*). The BMP pathway is activated by BMP ligands binding to BMP receptors, which can then phosphorylate SMAD1, SMAD5, and SMAD8 (also called SMAD9). Phosphorylated SMAD1/5/8 associates with co-SMAD4, forming a complex that can translocate to the nucleus and regulate expression of target genes. BMP signaling is important in early embryonic dorsoventral patterning and induction of the neural crest (*Garnett et al., 2012*; *Hashiguchi and Mullins, 2013*; *McMahon et al., 1998*; *Schumacher et al., 2011*). Following neural crest induction, BMP signaling has been implicated in patterning within the neural crest and surrounding tissues, as well as development of nervous system- and musculoskeletal-related neural crest lineages (*Hayano et al., 2015*; *McMahon et al., 1998*; *Nikaido et al., 1997*; *Reichert et al., 2013*; *Valdivia et al., 2016*). While many developmental functions of BMP signaling are well characterized, the relationship of BMP signaling to the development of pigment cells from the neural crest is poorly understood.

Our laboratory recently identified a BMP ligand, *GDF6,* that acts to suppress differentiation and cell death in melanoma (*Venkatesan et al., 2018*). We found that *GDF6*-activated BMP signaling in melanoma cells represses expression of *MITF*, a key regulator of melanocyte differentiation, leading to a less differentiated state. Here, we investigate the role of *GDF6* and the BMP pathway in development of pigment cells in zebrafish. We show that BMP signaling regulates fate specification of neural crest-derived pigment cell lineages and suppresses expression of *mitfa*, an ortholog of *MITF*. Furthermore, we show that disrupting BMP signaling alters fate specification between melanocyte and iridophore populations in the zebrafish. We determine that this shift in fate occurs at the level of an *mitfa*-positive pigment progenitor cell, and that BMP signaling acts through *mitfa* to direct *mitfa*-positive pigment progenitor cells to a specific fate. Altogether, these findings suggest pathologic BMP signaling in melanoma is a reiteration of normal physiologic function of BMP signaling during melanocyte development.

## Results

### Loss of *gdf6a* leads to an increase in adult pigmentation

To understand potential functions of *gdf6a* in the melanocyte lineage, we first determined if any alterations in pigment pattern were present in animals lacking *gdf6a*. In these studies, we used the *gdf6a*[s327] allele, hereafter referred to as *gdf6a(lf),* which encodes an early stop codon and has previously been shown to cause a complete loss of *gdf6a* function (*Gosse and Baier, 2009*). Previous studies have identified early roles for *gdf6a* during initial embryonic patterning, including dorsoventral patterning immediately following fertilization, thus *gdf6a(lf)* mutants have significantly decreased viability during the first 5 days post fertilization (*Sidi et al., 2003*). However, we found that a small proportion of *gdf6a(lf)* animals are able to survive early development and progress to adulthood. These *gdf6a(lf)* adult zebrafish had increased pigmentation when compared to wild-type zebrafish (*Figure 1A*). Furthermore, *gdf6a(lf)* adult zebrafish had qualitative disruption of the normal pigment pattern of both stripe and scale-associated melanocytes, and a significant increase in the number of scale-associated melanocytes as well as the overall scale area covered by melanin (*Figure 1A and B*). These results indicate that *gdf6a(lf)* mutants have melanocyte defects.

### Loss of *gdf6a* or inhibition of BMP signaling leads to an increase in embryonic melanocytes

Since zebrafish develop their adult pigment pattern during metamorphosis, it is possible *gdf6a* acts during this stage to change adult pigmentation, and not during initial pigment cell development in embryogenesis (*Parichy and Spiewak, 2015*; *Patterson and Parichy, 2013*; *Quigley et al., 2004*).

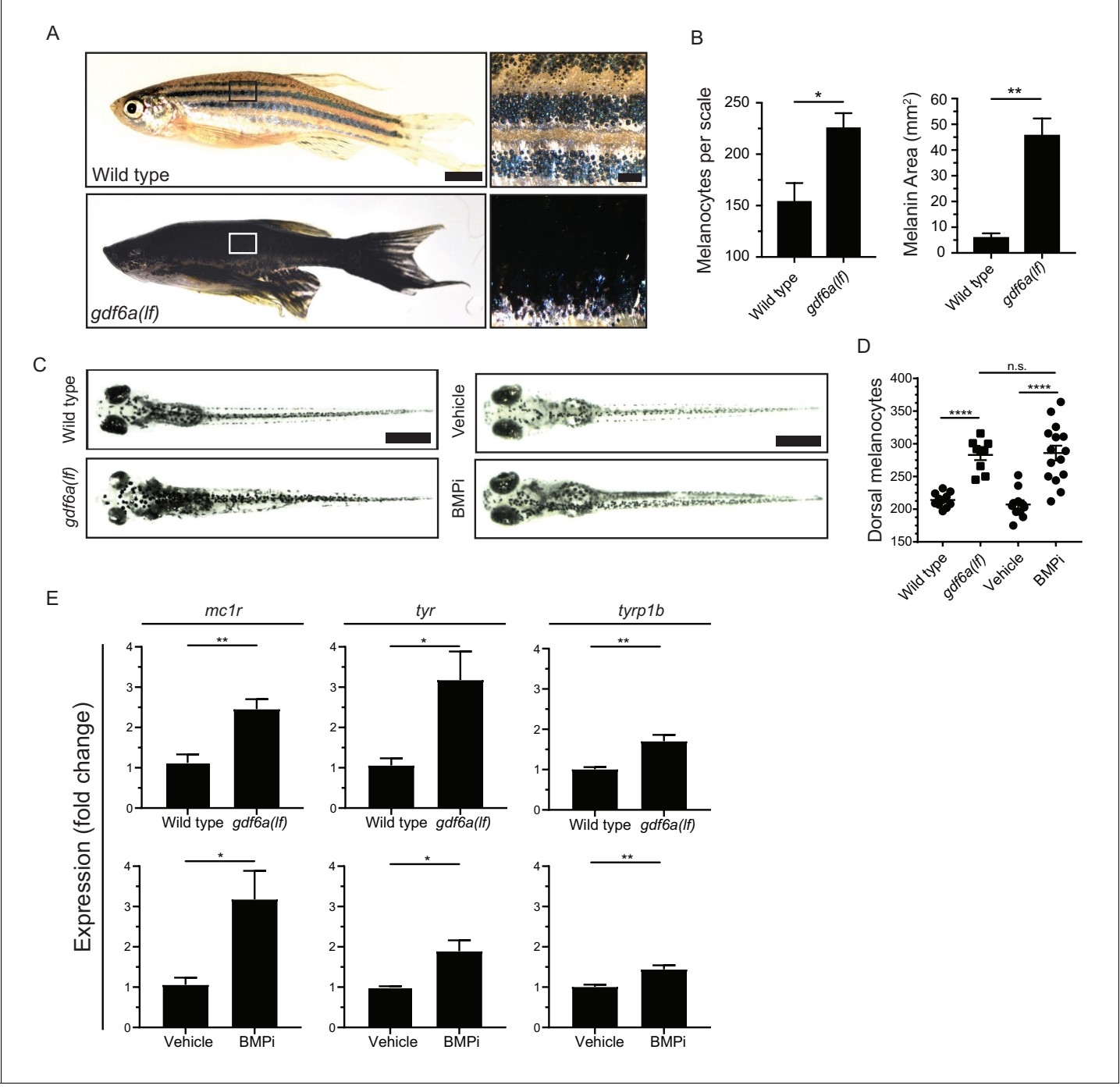

**Figure 1.** *gdf6a* loss or BMP inhibition causes the development of supernumerary melanocytes. (**A**) Images of wild-type and *gdf6a(lf)* adult zebrafish, scale bar = 4 mm, inset scale bar = 1 mm. (**B**) Quantification of number of melanocytes (left) and scale pigmentation using melanin coverage (right), n = 3 scales per group. (**C**) Wild-type and *gdf6a(lf)* embryos imaged at 5 days post fertilization (DPF); vehicle- and BMPi-treated embryos imaged at 5 DPF. Scale bar = 1 mm. Animals were treated with epinephrine prior to imaging. (**D**) Quantification of dorsal melanocytes per animal in 5 DPF wild-type, *gdf6a(lf)* mutant, vehicle-, and BMPi-treated embryos. n = 11, 9, 11, and 15 embryos, respectively, from two independent experiments (N = 2). (**E**) Expression of melanocyte differentiation markers *mc1r, tyr,* and *tyrp1b* by qRT-PCR in wild-type, *gdf6a(lf)* mutant, vehicle-, and BMPi-treated embryos. n = 5–6 replicates across two independent experiments (N = 2) for each group. Expression was normalized to *β-actin*. Error bars represent mean + /- SEM. P-values were calculated using Student's t-test in panel B and E, and one-way ANOVA with Tukey's multiple comparisons test in panels D, *p<0.05, **p<0.01, ***p<0.001, ****p<0.0001, n.s., not significant.

The online version of this article includes the following source data and figure supplement(s) for figure 1:

**Source data 1.** *gdf6a* loss or BMP inhibition causes the development of supernumerary melanocytes.

*Figure 1 continued on next page*

*Figure 1 continued*

**Figure supplement 1.** *gdf6* paralogs are necessary for normal embryonic development.

To address this issue, we investigated whether *gdf6a(lf)* caused embryonic pigmentation changes and, if so, whether any such changes were BMP-dependent. We crossed *gdf6a(lf)* heterozygotes and, in randomly selected progeny, quantified the number of melanocytes that developed by 5 days post-fertilization (DPF). Following melanocyte quantification, we determined the genotype of each embryo. In parallel, we treated wild-type zebrafish during the period of neural crest induction and melanocyte specification (12 to 24 hours post fertilization) with a small molecule BMP inhibitor, DMH1, hereafter referred to as BMPi, and performed the same quantification of embryonic melanocytes (*Hao et al., 2010*). *gdf6a(lf)* homozygous animals developed approximately 40% more dorsal melanocytes by 5 DPF, when compared to sibling wild-type animals and *gdf6a(lf)* heterozygotes (*Figure 1C and D*, *Figure 1—figure supplement 1A*). *gdf6a(lf)* animals also showed increased expression of *mc1r*, *tyr*, and *tyrp1b*, all markers of differentiated melanocytes, which is consistent with an increase in melanocyte number (*Figure 1E*). Furthermore, treatment with BMPi phenocopied the melanocyte changes observed in *gdf6a(lf)* mutants, coupled with a similar increase in expression of *mc1r*, *tyr*, and *tyrp1b* (*Figure 1D and E*). We observed a similar increase in total body melanocytes, indicating that there is an overall increase in melanocyte development instead of a failure of migration leading to a specific increase in dorsal melanocytes (*Figure 1—figure supplement 1B*). These results indicate *gdf6a*-activated BMP signaling normally acts in embryos to limit melanocyte development.

## *gdf6* ortholog expression during neural crest development

Numerous BMP ligands are expressed during early embryogenesis and participate in multiple facets of development, including neural crest induction. It was previously shown that multiple BMP ligands are activated during zebrafish neural crest development (*Reichert et al., 2013*). Of those ligands investigated, only *gdf6a* and *bmp6* were expressed in the neural crest, and only *gdf6a* activated BMP signaling within neural crest cells. An additional study identified dorsal expression of a zebrafish paralog of *gdf6a*, *gdf6b*, indicating it could potentially act in the neural crest (*Bruneau and Rosa, 1997*). We verified *gdf6b* expression is restricted to the neural tube, and further determined *gdf6b* loss of function has no impact on pigment cell development by generating a *gdf6b* mutant, hereafter referred to as *gdf6b(lf)*, and counting embryonic melanocytes (*Figure 1—figure supplement 1D–G*). We generated double mutants for both *gdf6a(lf)* and *gdf6b(lf)* to assess whether these paralogs functioned redundantly or could compensate for the loss of one another. Unfortunately, *gdf6a(lf); gdf6b(lf)* double mutants had significant morphologic defects and decreased viability such that we could not adequately compare melanocyte numbers in these animals (*Figure 1—figure supplement 1H–I*). However, because there were no pigmentation defects in *gdf6b(lf)* mutants and *gdf6a(lf)* pigmentation defects were the same severity as observed in animals treated with a pan-BMP inhibitor, it is likely that most, if not all, effects of BMP signaling on melanocyte development are directed by *gdf6a*.

## BMP inhibition increases *mitfa*-positive pigment cell progenitors in the neural crest

We sought to determine the mechanism by which BMP signaling inhibits melanocyte development in embryos. Based on our experiments using BMPi, we suspected BMP signaling acts during pigment cell development from the neural crest to prevent an increase in melanocytes. Following induction, neural crest cells undergo proliferation, followed by fate restriction and specification, in which individual cells become less and less multipotent until a single possible fate remains (*Jin et al., 2001*; *Lewis, 2004*; *Nagao et al., 2018*). In many cases, specification to the ultimate lineage is determined by activation of an individual or a group of lineage-specific factors (*Sauka-Spengler et al., 2007*). For pigment cells, fate specification is dependent on integration of many signaling factors, including BMP and Wnt signaling, as well as key transcription factors, such as AP2α, AP2ε, SOX-, PAX-, and FOX-family transcription factors (*Garnett et al., 2012*; *Ignatius et al., 2008*; *Lister et al., 2006*; *Sato et al., 2005*; *Southard-Smith et al., 1998*; *Thomas and Erickson, 2009*; *Van Otterloo et al.,*

*2010*). In zebrafish, specification of the melanocyte lineage depends on upregulation of *sox10* and downregulation of factors inhibiting differentiation, such as *foxd3* (*Curran et al., 2010*; *Curran et al., 2009*; *Dutton et al., 2001*). Following *sox10* upregulation, a subset of *sox10*-positive cells can activate pigment lineage markers associated with melanocytes, iridophores, and xantho-phores (*Elworthy et al., 2003*; *Fadeev et al., 2016*; *Nagao et al., 2018*; *Nord et al., 2016*; *Petratou et al., 2018*). *mitfa* is a key factor that is expressed early in pigment progenitor cells (*Lister et al., 1999*). Based on this framework, we hypothesized two potential mechanisms by which supernumerary melanocytes are generated: 1) an increase in proliferation of either neural crest cells or pigment progenitor cells, or 2) an increase in the proportion of neural crest cells that are specified to become pigment progenitor cells. To assess changes in proliferation of neural crest cells and pig-ment cells, we analyzed cell cycle profiles using flow cytometry. Embryos expressing reporters for neural crest cells (Tg(crestin:eGFP)) or pigment progenitor cells (Tg(*mitfa:eGFP*)) were treated with BMPi from 12 to 24 HPF, during neural crest development and specification (*Curran et al., 2009*; *Kaufman et al., 2016*). Embryos were dissociated, stained with DAPI, and analyzed for DNA content of neural crest cells or pigment progenitor cells as defined by the fluorescent GFP marker (*Fig-ure 2—figure supplement 1A*). We observed no increase in the percent of S/G2/M cells in either population, indicating no apparent change to cell cycle distribution of either neural crest cells or pig-ment progenitor cells (*Figure 2—figure supplement 1B–C*). To verify these findings reflected no change in proliferation rate, we performed a 5-ethynyl-2'-deoxyuridine (EdU) incorporation assay. We treated *Tg(crestin:eGFP)* or *Tg(mitfa:eGFP)* embryos with BMPi or vehicle control along with EdU during early (12–14 HPF), middle (16–18 HPF), and late (20–22 HPF) stages of neural crest and pigment cell development (*Figure 2—figure supplement 1D–E*). We observed no differences in EdU incorporation between BMPi-treated and vehicle control groups, indicating no change in prolif-eration rates of either *crestin:eGFP*-positive neural crest cells or *mitfa:eGFP*-positive pigment pro-genitor cells (*Figure 2—figure supplement 1F*). Without an obvious increase in proliferation, we tested the hypothesis that a change in specification results in increased melanocytes. To assess changes in specification of neural crest cells into pigment progenitor cells, we utilized reporter embryos marking neural crest cells in red (*Tg(crestin:mCherry)*) and pigment progenitor cells in green (*Tg(mitfa:eGFP)*) (*Figure 2A*). Using these reporters, neural crest cells not committed to the pigment cell lineage are *crestin:mCherry* single-positive, whereas *crestin:mCherry/mitfa:eGFP* dou-ble-positive cells are those newly committed to the pigment cell lineage. We treated embryos con-taining both reporter transgenes with BMPi from 12 to 24 HPF, during neural crest development and specification. At 24 HPF, we dissociated embryos and analyzed cells for fluorescent marker expression by flow cytometry (*Figure 2A*). Embryos treated with BMPi showed approximately a 1.5-fold increase in the percentage of *crestin:mCherry/mitfa:eGFP* double-positive cells per total *crestin:mCherry*-positive cells (*Figure 2B and C*). We further verified a change in specification by staining BMPi- or vehicle-treated *Tg(crestin:eGFP)* embryos with anti-Mitfa antibody and assessed the pro-portion of *crestin:eGFP*-positive cells that stained positive for Mitfa (*Figure 2D*). We observed a 1.3-fold increase in the proportion of Mitfa/*crestin:eGFP* double-positive cells per total *crestin:eGFP*-positive cells in animals treated with BMPi compared to vehicle control (*Figure 2E*). Altogether these results suggest that an increase in embryonic melanocytes is caused by an increase in the proportion of neural crest cells specified as pigment progenitor cells, rather than a change in proliferation of either neural crest or pigment progenitor cells.

## BMP signaling in *mitfa*-expressing pigment progenitor cells can alter melanocyte development in embryogenesis

Because we observed an impact of BMP signaling on neural crest-to-pigment progenitor cell specifi-cation, we explored the relationship between *gdf6a* and *mitfa* expression. First, using whole-mount in situ hybridization (*Figure 3A*), we found that the anteroposterior expression domains of *gdf6a* and *mitfa* were mostly, if not completely, non-overlapping. As described previously (*Reichert et al., 2013*; *Rissi et al., 1995*), *gdf6a* was expressed in the anterior half of the embryo in the neural crest at 12 HPF, shortly following its induction. Consistent with previous observations (*Lister et al., 1999*), *mitfa* was not expressed at this time. At 18 HPF, *gdf6a* was absent from the anterior neural crest, but instead was restricted to the posterior half of the embryo with expression apparent in the hypo-chord and epidermal cells (*Rissi et al., 1995*). At 24 HPF, *gdf6a* expression was further restricted to

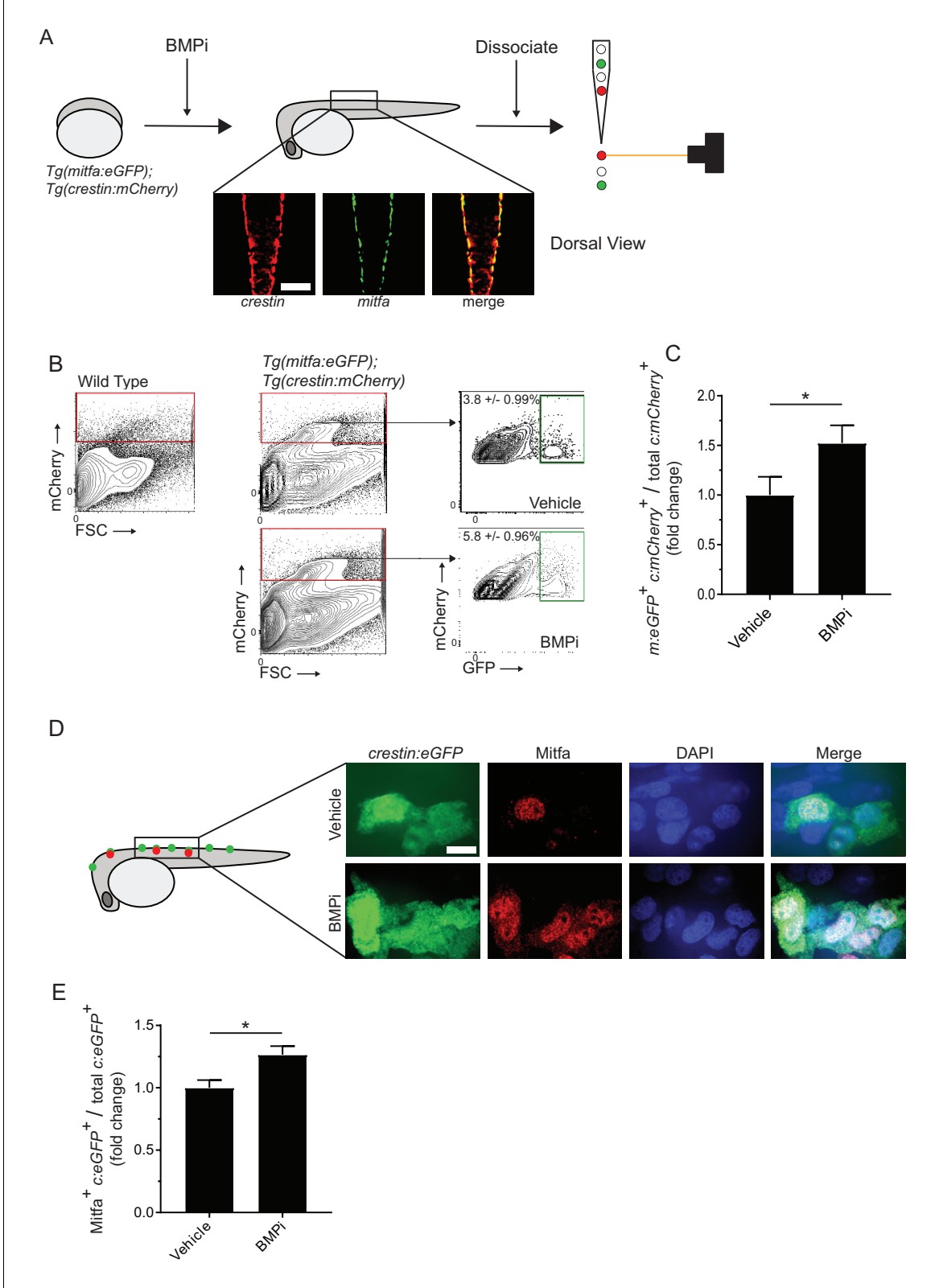

**Figure 2.** Inhibition of BMP signaling increases *mitfa*-positive neural crest cells. (A) Diagram of experiment. *Tg(crestin:mCherry); Tg(mitfa:eGFP)* embryos were treated with BMPi from 12 to 24 HPF. At 24 HPF, embryos were dissociated and analyzed via flow cytometry for GFP- and mCherry-positive cells, scale bar = 200 μm. (B) Gating strategy based on non-transgenic wild-type control to identify *crestin:mCherry*-positive cells and *crestin: mCherry/mitfa:eGFP* double-positive cells. Top, control vehicle-treated embryos. Bottom, BMPi-treated embryos. (C) Fold change in *crestin:mCherry/*

*Figure 2 continued on next page*

Figure 2 continued

*mitfa:eGFP* double-positive cells per total *crestin:mCherry*-positive cells in vehicle and BMPi-treated groups, N = 3 biological replicates of 80–100 stage-matched embryos pooled for each condition. m:eGFP, *mitfa:eGFP*; c:mCherry, *crestin:mCherry*. (D) anti-Mitfa immunofluorescence in *Tg(crestin: eGFP)* embryos treated with BMPi or vehicle control and fixed at 24 hr, scaled bar = 10 μm. (E) Fold change in Mitfa/*crestin:eGFP* double-positive cells per total *crestin:eGFP*-cells, n = 16 embryos from two independent experiments (N = 2) for each condition. c:eGFP, *crestin:eGFP*. Error bars represent mean + /- SEM; P-value was calculated using ratio-paired t-test in panel C and Student's t-test in panel E, *p<0.05.

The online version of this article includes the following source data and figure supplement(s) for figure 2:

**Source data 1.** Inhibition of BMP signaling increases *mitfa*-positive neural crest cells.

**Figure supplement 1.** Increased proliferation is not observed in neural crest and pigment progenitor cell populations of BMPi-treated embryos.

the posterior end of the embryo and *mitfa* expression expanded posteriorly to a commensurate degree.

To quantify changes in expression of *gdf6a* and *mitfa* specifically in neural crest cells, we isolated eGFP-positive cells from *Tg(crestin:eGFP)* embryos at 16 HPF and 22 HPF by FACS. We analyzed the relative levels of *gdf6a* and *mitfa* expression using qRT-PCR and found that *gdf6a* expression is relatively higher than *mitfa* at 16 HPF, while *mitfa* expression is relatively higher at 22 HPF, indicating an inverse correlation of expression over the course of neural crest development (*Figure 3B*). The reciprocal nature of *gdf6a* and *mitfa* expression changes is consistent with the possibility that *gdf6a*-driven BMP signaling acts in neural crest cells to repress *mitfa* expression and prevent excess pigment progenitor cells from being specified. However, we also considered the possibility that BMP signaling is active in *mitfa*-positive cells and affects the fates of these cells. To determine if BMP signaling is active in *mitfa*-positive cells, we stained *Tg(mitfa:eGFP)* zebrafish with antibodies against phosphorylated-SMAD-1/5/8 (pSMAD). We verified specificity of the anti-pSMAD antibody using BMPi treated embryos (*Figure 3—figure supplement 1*). 30% of *mitfa*-expressing cells on the leading, posterior edge of the *mitfa* expression domain had nuclear-localized pSMAD staining, whereas only 7% of *mitfa*-expressing cells in regions anterior to the leading edge showed nuclear pSMAD staining (*Figure 3C and D*). These results suggest BMP signaling is active as *mitfa*-expressing cells first arise in the neural crest, but is turned off in such cells as development proceeds.

To assess if BMP activity in *mitfa*-expressing cells can impact melanocyte development, we directly altered BMP activity in these cells. We first generated a stably transgenic zebrafish line expressing *gdf6a* under the control of the *mitfa* promoter (*Tg(mitfa:gdf6a)*) to increase *gdf6a* expression in *mitfa*-expressing cells. Embryos expressing the *Tg(mitfa:gdf6a)* transgene developed fewer melanocytes than non-transgenic sibling controls (*Figure 4A*). To alter BMP signaling in a cell-autonomous manner within *mitfa*-expressing cells, we used the miniCoopR system in two complementary approaches: a) to express a dominant negative BMP receptor (dnBMPR), which suppresses intracellular BMP activity, and b) to express a phospho-mimetic variant of SMAD1 (SMAD1-DVD) to constitutively activate intracellular BMP activity (*Ceol et al., 2011*; *Nojima et al., 2010*; *Pyati et al., 2005*). We injected *mitfa(lf)* animals with miniCoopR-dnBMPR, miniCoopR-SMAD1-DVD, or control miniCoopR-eGFP (*Figure 4B*). At 5 DPF, we scored animals for rescue of melanocytes. Animals injected with miniCoopR-dnBMPR showed a rescue rate of 79% as compared to 29% of miniCoopR-eGFP-injected animals. Furthermore, animals injected with miniCoopR-SMAD1-DVD showed a 15% rescue rate (*Figure 4C*). Together these results suggest BMP signaling is active in *mitfa*-expressing cells and modulating BMP signaling can alter the fate of these *mitfa*-expressing cells during development. Thus, *gdf6a*-driven BMP signaling can both limit the number of *mitfa*-expressing cells arising from the neural crest but also act in *mitfa*-expressing pigment progenitor cells to influence their development into melanocytes.

## Iridophores, but not other neural crest derivatives, are reduced upon *gdf6a* loss

Because we observed no change in proliferation of *crestin*- or *mitfa*-positive populations, but the number of melanocytes developing from these precursors was increased, we questioned whether this increase corresponded with a commensurate loss of a related pigment or other neural crest-derived cell type. To determine what cells may be impacted, we looked for transcriptional changes in markers of other, related neural crest derivatives as well as known neural crest factors important in

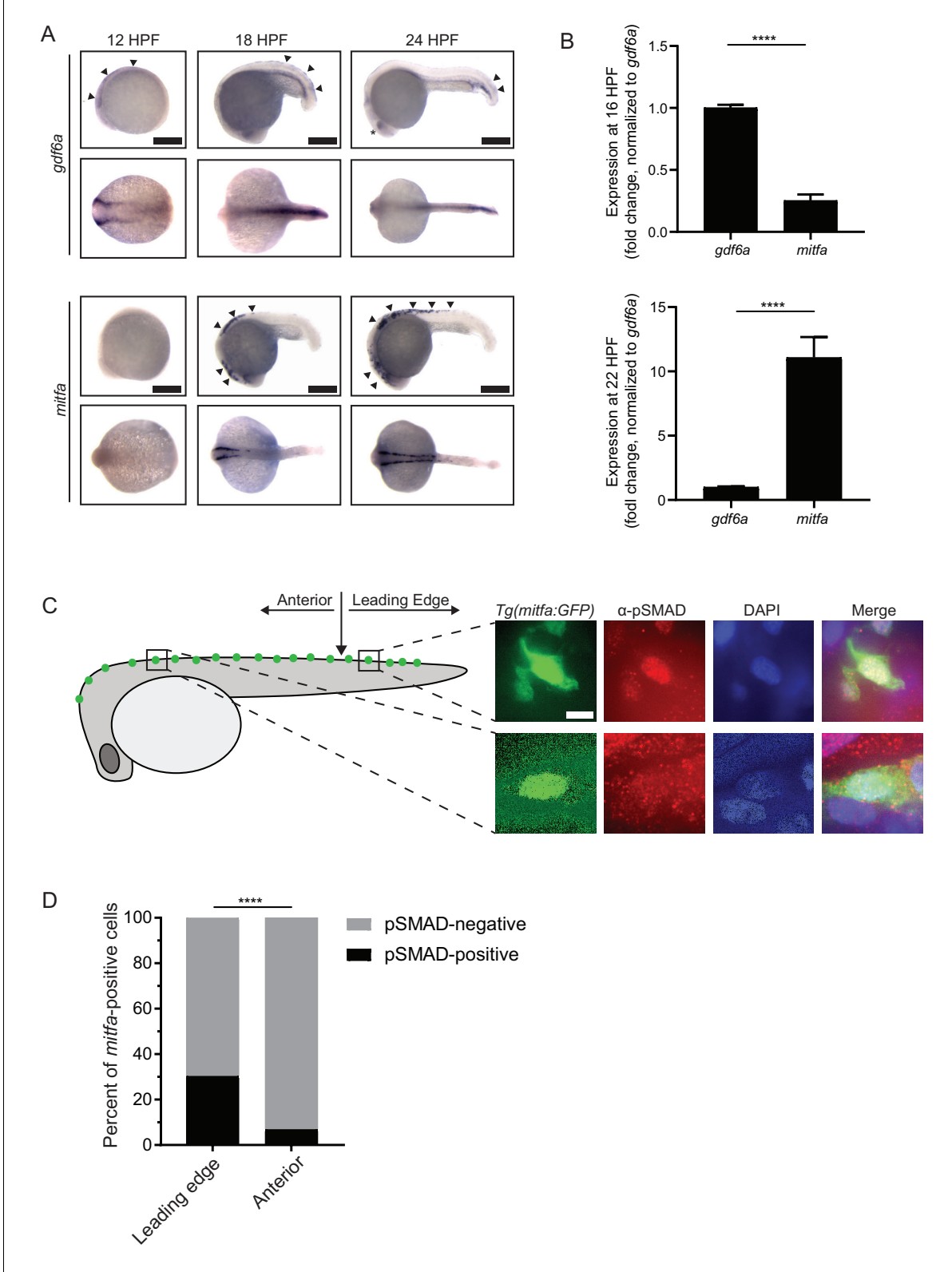

**Figure 3.** *gdf6a* expression and BMP activity in pigment progenitor cells. (**A**) RNA in situ hybridization for *gdf6a* (top) and *mitfa* (bottom) at 12-, 18-, and 24 hr post-fertilization. Arrowheads indicate expression domains in the region of the neural crest of *gdf6a* and *mitfa*. Asterisk indicates known dorsal retinal expression of *gdf6a*. Scale bar = 500 μm. (**B**) Expression of *gdf6a* and *mitfa* from neural crest cells isolated from *Tg(crestin:eGFP)* embryos by FACS at 16 HPF and 22 HPF. Samples were normalized to *gdf6a* expression. n = 5–6 replicates per conditions from two independent experiments

*Figure 3 continued on next page*

*Figure 3 continued*

(N = 2). (**C**) Images of GFP-positive cells from *Tg(mitfa:eGFP)* zebrafish stained with α-pSMAD 1/5/8 antibody. Scale bar = 10 μm. (**D**) Quantification of *mitfa:eGFP*-positive cells that are phospho-SMAD1/5/8-positive. The leading edge encompassed the five most posterior *mitfa*-positive cells, whereas anterior cells constituted any *mitfa*-positive cells anterior to the leading edge. n = 102 and 186 for distal leading edge and anterior cells, respectively, from three independent experiments (N = 3). P-values were calculated using Fisher's exact test for panels B and D, ****p<0.0001.
The online version of this article includes the following source data and figure supplement(s) for figure 3:

**Source data 1.** *gdf6a* expression and BMP activity in pigment progenitor cells.
**Figure supplement 1.** Treatment with the BMP inhibitor DMH1 reduces phospho-SMAD1/5/8 staining in embryos Top, vehicle-treated animals and, bottom, BMPi-treated animals showing lateral views of developing muscle segments, identified by asterisks.

specifying different cell fates. We isolated RNA from *gdf6a(lf)* and wild-type embryos at 5 DPF. Additionally, we isolated RNA from embryos treated with a BMPi or vehicle control. We performed qRT-PCR for markers of neural crest derivatives, including *mbpa* for glial cells, *pomca* for adrenal medullary cells, *neurog1* for neuronal cells, *aox5* for xanthophores, and *pnp4a* for iridophores, as well as *sox10* and *foxd3* (*Fadeev et al., 2016*; *McGraw et al., 2008*; *Parichy et al., 2000*; *Thomas and Erickson, 2009*). As a control, we used a chondrocyte marker, *col2a1a*, as craniofacial development has previously been described to be disrupted by *gdf6a* loss (*Reed and Mortlock, 2010*). Per our previous analysis, *gdf6a(lf)* mutants and BMPi-treated embryos demonstrated an increase in expression of the melanocyte markers *mc1r*, *tyr*, and *tyrp1b* (*Figure 1E*). And as predicted based on previous literature, *gdf6a(lf)* mutants and BMPi-treated embryos showed a decrease in expression of the chondrocyte marker, *col2a1a*. We observed no changes in *foxd3* expression, but observed a slight downregulation of *sox10* (*Figure 5—figure supplement 1A*), previously shown to be downregulated upon *GDF6* knockdown in melanoma cells (*Venkatesan et al., 2018*). Because *foxd3* has previously been shown to be related to both BMP signaling in neural crest induction and pigment cell development (*Curran et al., 2010*; *Curran et al., 2009*; *Stewart et al., 2006*), we questioned if *foxd3* may have a functional role in mediating the effect of BMP signaling on melanocyte development despite no change in expression. We treated *foxd3(lf)* (*Stewart et al., 2006*) embryos with BMPi and assessed melanocyte development (*Figure 5—figure supplement 1B*). We observed a comparable increase in melanocytes in *foxd3(lf)* and sibling control animals, indicating BMP signaling acts independently of *foxd3* to alter melanocyte development. Despite a downregulation of *sox10* in *gdf6a* mutants and BMPi-treated embryos, we still observed an increase in *mitfa* expression and markers of melanocyte differentiation, and an increase in the number of melanocytes.

In our evaluation of neural crest derivative populations, markers for neuronal, glial, adrenal medullary, and xanthophore lineages were no different in *gdf6a* compared to wild-type animals (*Figure 5A*). Similar results were obtained in animals treated with a BMPi, with the exception of a change in *mbpa* expression, a marker for glial cells. Previous studies have shown glial cell development is regulated in part by BMP activity (*Jin et al., 2001*). Since *mbpa* expression was unchanged in *gdf6a(lf)* animals, this suggests another BMP ligand is involved in activating BMP signaling to promote glial cell development. For neuronal and xanthophore cell populations, we verified that the expression profile correlated with cell numbers or development of key structures. We treated animals with BMPi or vehicle and stained with anti-HuC/D antibody to label neuronal cells in the dorsal root ganglia and developing gastrointestinal tract (*Lister et al., 2006*) (*Figure 5—figure supplement 1C–E*). We detected no difference in dorsal root ganglia and enteric neuron development between each group. We imaged animals stably expressing *Tg(aox5:PALM-eGFP)* to label xanthophores and found no qualitative difference in xanthophores between BMPi- and vehicle-treated groups (*Eom and Parichy, 2017*) (*Figure 5—figure supplement 1F*). We further evaluated *mitfa*-positive cells associated with the dorsal root ganglion that have previously been connected to formation of adult pigment and are proposed to be melanocyte stem cells (*Dooley et al., 2013*; *Hultman et al., 2009*; *Singh et al., 2016*). Using *Tg(mitfa:eGFP)* embryos, we observed an increase in these DRG-associated, *mitfa:eGFP*-positive cells, suggesting that BMPi not only increases differentiated embryonic melanocytes but also cells that underlie adult pigmentation (*Figure 5—figure supplement 1G*).

In our transcriptional analyses of *gdf6a(lf)* and BMPi-treated embryos, we observed a decrease in expression of *pnp4a*, a marker for the iridophore lineage, indicating a potential deficit of iridophore development (*Figure 5A*). Since *pnp4a* is expressed in other developing cells and tissues, such as

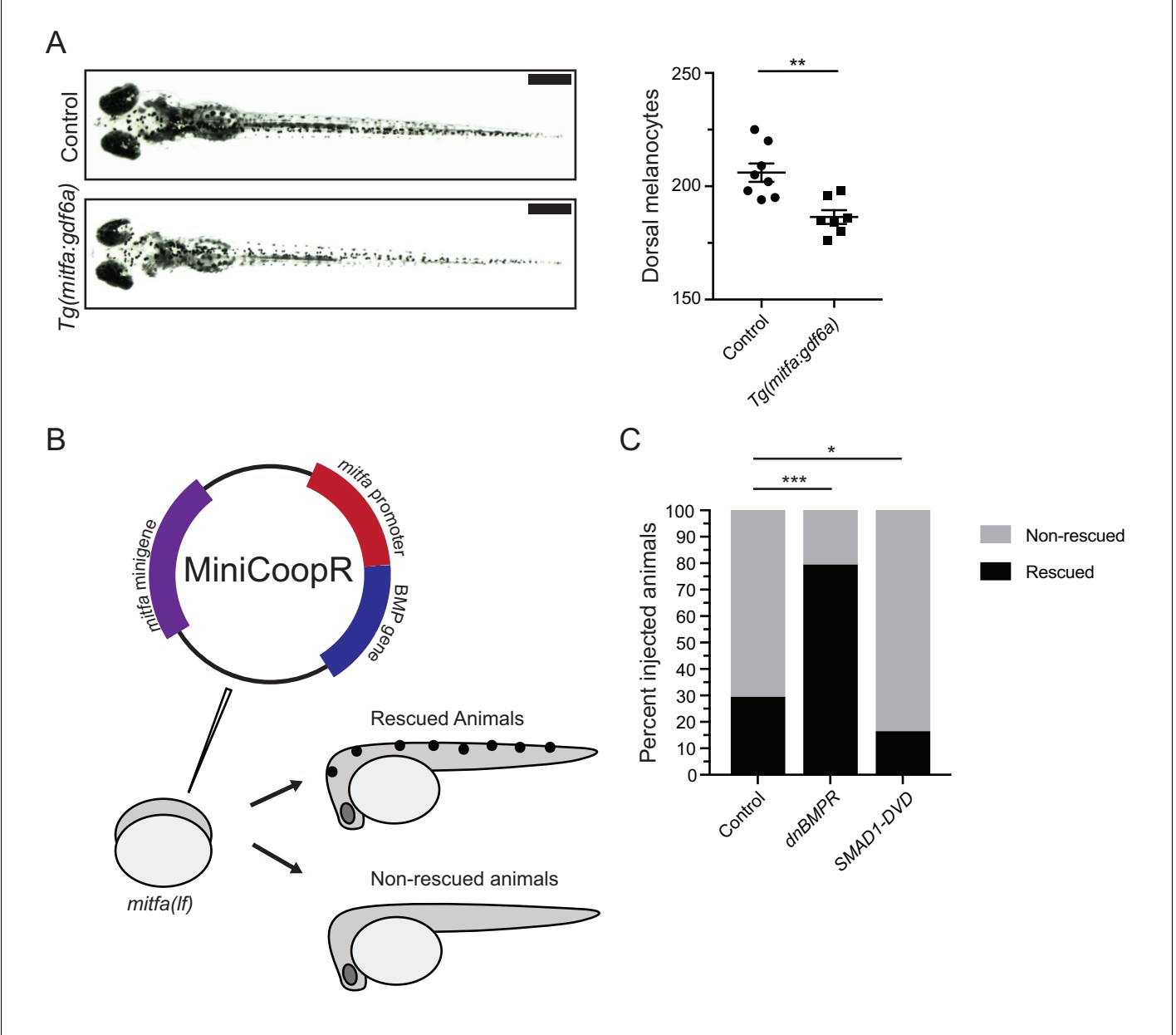

**Figure 4.** BMP signaling within pigment progenitor cells can impact embryonic melanocytes. (**A**) *Tg(mitfa:gdf6a)* and non-transgenic sibling control embryos (left), and quantification of dorsal melanocytes per animal in each group (right). Animals were treated with epinephrine prior to imaging at 5 DPF, n = 8 and 7 for control and *Tg(mitfa:gdf6a)* groups, respectively, from two independent experiments (N = 2). Scale bar = 1 mm. (**B**) Diagram of miniCoopR rescue experiment. Animals harboring a *mitfa(lf)* mutation were injected at the single-cell stage with the miniCoopR vector containing a BMP gene. Animals were evaluated at 5 DPF for the presence of melanocytes. If melanocytes were present, that animal was scored as rescued, whereas animals lacking melanocytes were scored as non-rescued. (**C**) Percentages of rescued and non-rescued animals following injection of a miniCoopR-BMP vector, n = 361, 193 and 152 for control, *dnBMPR*, and *SMAD1-DVD* groups, respectively, from four independent experiments (N = 4). Error bars represent mean + /- SEM. P-values were calculated Student's t-test for panel A and with Fisher's exact test with Bonferroni's correction for panel C, *p<0.05, **p<0.01, ***p<0.001.

The online version of this article includes the following source data for figure 4:

**Source data 1.** BMP signaling within pigment progenitor cells can impact embryonic melanocytes.

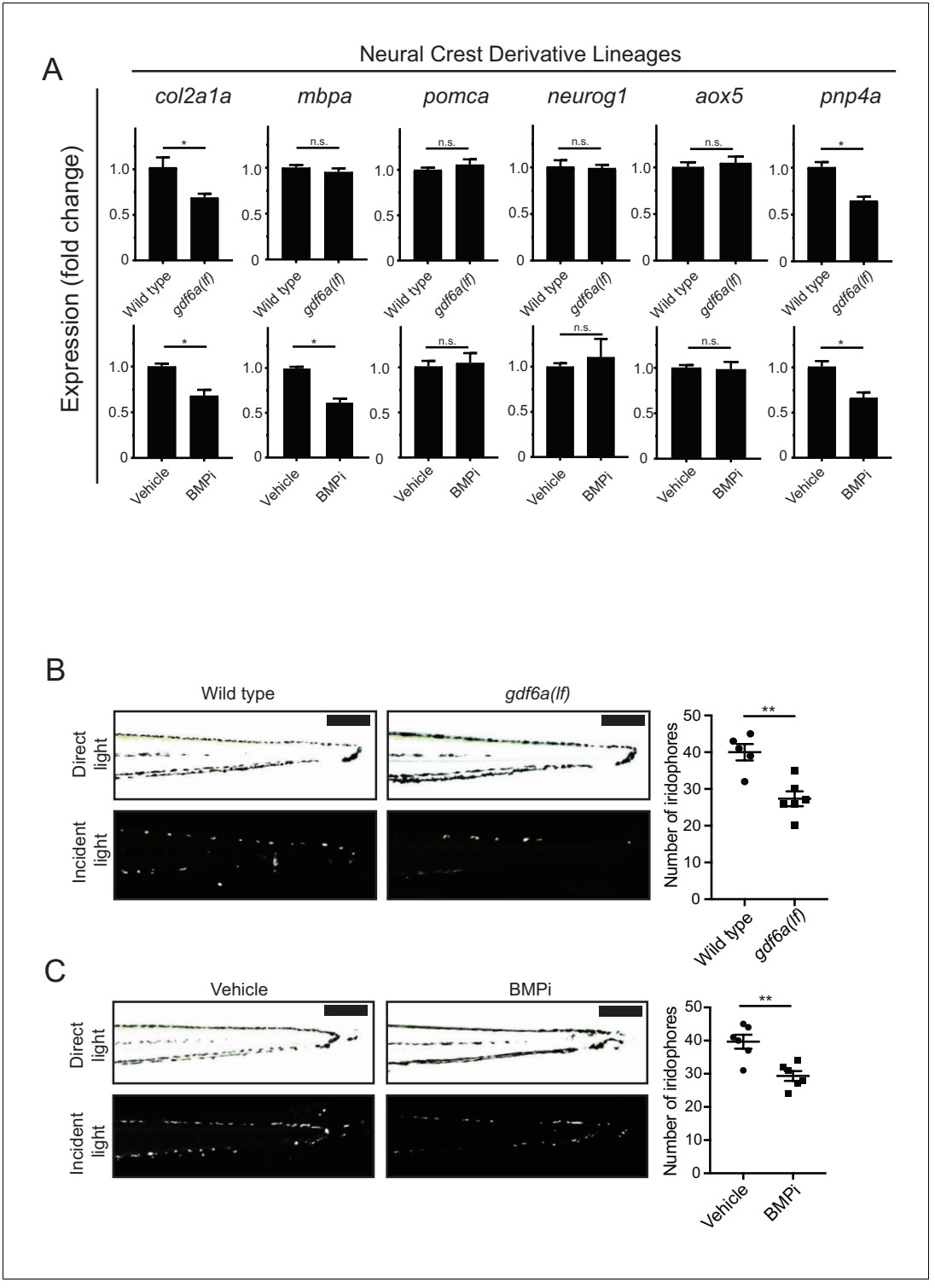

**Figure 5.** *gdf6a* loss and BMP inhibition impact development of specific neural crest derivatives. (**A**) Expression analyses of multiple neural crest and neural crest derivative lineage markers. qRT-PCR was used to assess changes in markers of neural crest markers and neural crest derivatives in *gdf6a(lf)* embryonic zebrafish (top) and BMPi-treated wild-type zebrafish (bottom) at 5 DPF; *col2a1a,* chondrocytes; *mbpa,* glial; *pomca,* adrenal medullary cells; *neurog1,* neuronal cells; *aox5,* xanthophores; *pnp4a,* iridophores; n = 5–6 for each group from two independent experiments (N = 2). (**B**) Direct light (top) and incident light (bottom) images of wild-type and *gdf6a(lf)* embryos at 5 DPF and quantification of dorsal iridophores (right) per animal in each group. Animals were treated with epinephrine prior to imaging at 5 DPF; n = 5 and 6 for wild-type and *gdf6a(lf)* groups, respectively, from two

*Figure 5 continued on next page*

*Figure 5 continued*

independent experiments (N = 2); scale bar = 500 µm. (C) Direct light, top, and incident light, bottom, images of wild-type embryos treated with vehicle or BMPi from 12 to 24 HPF and quantification of dorsal iridophores, right, per animal in vehicle and BMPi treated groups. Animals treated with epinephrine prior to imaging at 5 DPF, n = 6 and 6 for vehicle and BMPi groups, respectively, from two independent experiments (N = 2); scale bar = 1 mm. Error bars represent mean +/- SEM, P-values calculated with Student's t-test, *p<0.05, **p<0.01, n.s., not significant.

The online version of this article includes the following source data and figure supplement(s) for figure 5:

**Source data 1.** *gdf6a* loss and BMP inhibition impact development of specific neural crest derivatives.
**Figure supplement 1.** Neural crest cells and derivative populations show variable response to BMP inhibition.

retinal cell populations, we wanted to confirm these changes were specific to a deficit in neural crest-derived body iridophores (*Cechmanek and McFarlane, 2017*; *Lopes et al., 2008*; *Petratou et al., 2018*). We quantified the number of dorsal iridophores that developed in *gdf6a(lf)* embryos (*Figure 5B*) and embryos treated with BMPi (*Figure 5C*) at 5 DPF, using incident light to highlight embryonic iridophores. Embryos developed 32% and 27% fewer iridophores with *gdf6a(lf)* or BMPi treatment, respectively. Together, these results indicate that *gdf6a*-driven BMP signaling promotes iridophore development.

## BMP inhibition increases the likelihood a multipotent precursor will develop into a melanocyte

Melanocytes and iridophores have previously been shown to develop from *mitfa*-expressing pigment progenitor cells (*Curran et al., 2010*; *Curran et al., 2009*). To determine if BMP signaling regulates fate specification of melanocytes and iridophores from *mitfa*-expressing pigment progenitor cells, we performed lineage tracing. We injected *Tg(ubi:switch)* embryos, which stably express a *ubi:loxp-GFP-STOP-loxp-mCherry-STOP* transgene (*Mosimann et al., 2011*) with a *mitfa:Cre-ERT2* transgene to generate mosaic expression of Cre-ERT2 in *mitfa*-positive cells (*Figure 6A*). Injected embryos were treated with BMPi and hydroxytamoxifen (4-OHT), the latter to allow nuclear localization of Cre and generate recombinant events in individual *mitfa*-expressing pigment progenitor cells. Since these *mitfa*-expressing pigment progenitor cells are transient, 4-OHT treatment was limited to 12 to 24 HPF, with thorough embryo water exchange to wash out the drug and prevent recombinant events after specification. At 5 DPF, embryos with individual recombinant events, indicated by single mCherry-positive cells, were evaluated for the fate of those cells. In animals treated with BMPi, we observed an increase in the ratio of labeled melanocytes to iridophores as compared to vehicle-treated controls (*Figure 6B*, *Figure 6—figure supplement 1*). This result suggests that BMP signaling normally promotes the development of *mitfa*-expressing pigment progenitor cells into iridophores at the expense of melanocytes.

## BMP signaling represses *mitfa* expression within neural crest and pigment progenitor cells

Previous studies have indicated that the expression level of *mitfa* within pigment progenitor cells is important in specifying a melanocyte versus iridophore fate (*Curran et al., 2010*; *Curran et al., 2009*). Cells with a higher level *mitfa* expression are more likely to become melanocytes, while those that downregulate *mitfa* are more likely to become iridophores. Since *gdf6a(lf)* and BMP-inhibited embryos have excess melanocytes and fewer iridophores, we hypothesized that this phenotype resulted from disrupted regulation of *mitfa* expression in these embryos. This hypothesis was driven, in part, by our previous data in human melanoma cells, in which knockdown of *GDF6* decreased phospho-SMAD1/5/8 binding at the *MITF* locus and increased *MITF* expression (*Venkatesan et al., 2018*). We first determined the potential for *mitfa* to be regulated in a similar manner as *MITF*. We first looked at the flanking regions of both the *MITF* and *mitfa* loci for orthologous genes (*Catchen et al., 2009*). We found many of the same orthologs present near both loci, indicating a syntenic relationship (*Figure 7—figure supplement 1A*). We did not find conservation in *mitfa* of the phospho-SMAD-binding region defined in mammalian cells. However, in *mitfa* we did identify phospho-SMAD binding motifs (GC-SBM) similar to the one present in the mammalian phospho-

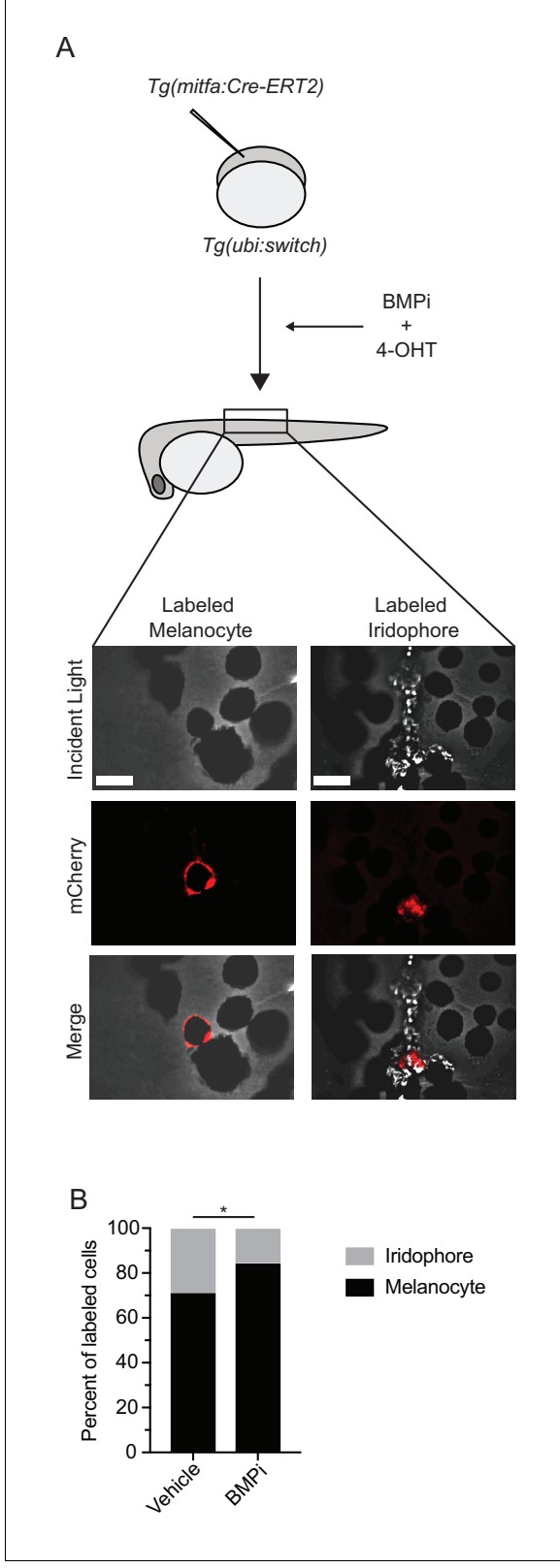

**Figure 6.** BMP inhibition impacts fate specification of *mitfa*-positive pigment progenitor cells. (**A**) Diagram of lineage tracing experiment. Embryos containing *Tg(ubi:switch)* were injected with a *mitfa:Cre-ERT2* construct and treated with BMPi and tamoxifen (4-OHT) from 12 to 24 HPF to block BMP signaling and allow Cre recombination. At 5 DPF, animals were screened for successful recombination by presence of single mCherry-labeled pigment

*Figure 6 continued on next page*

*Figure 6 continued*

cells, and the identities of those cells were assessed using incident light. Scale bar = 40 µm. (**B**) Quantification of mCherry-labeled cell fates at 5 DPF in vehicle and BMPi-treated animals, n = 101 and 80 labeled cells for vehicle and BMPi groups, respectively, from five independent experiments (N = 5); P-value calculated using Fisher's exact test, *p<0.05.

The online version of this article includes the following source data and figure supplement(s) for figure 6:

**Source data 1.** BMP inhibition impacts fate specification of *mitfa*-positive pigment progenitor cells.
**Figure supplement 1.** Quantification of iridophore and melanocyte numbers from lineage tracing Number of iridophores and melanocytes identified by lineage tracing under each condition.

SMAD-binding region (*Morikawa et al., 2011*), as well as additional phospho-SMAD-binding motifs that have been defined previously (*Jonk et al., 1998*). The presence of such sites suggests the potential for *mitfa* to be directly regulated by phospho-SMAD1/5/8 similarly to *MITF* (*Figure 7—figure supplement 1B*). To assess *mitfa* levels within neural crest cells and *mitfa*-expressing pigment progenitor cells, we treated *Tg(crestin:eGFP)* and *Tg(mitfa:eGFP)* embryos with BMPi as previously described. We dissociated embryos and used fluorescence-activated cell sorting (FACS) to isolate *crestin:eGFP*-positive or *mitfa:eGFP*-positive cells. We then assessed *mitfa* transcript levels in each population by qRT-PCR. Treatment with BMPi led to approximately 3-fold and 6-fold increases in *mitfa* expression in *crestin:eGFP*-positive and *mitfa:eGFP*-positive cells, respectively (*Figure 7A*). To explore this question on a single-cell level and analyze Mitfa protein levels, we stained BMPi-treated and vehicle-treated *Tg(crestin:eGFP)* embryos with an anti-Mitfa antibody (*Figure 7B*) (*Venkatesan et al., 2018*). In BMPi-treated animals, we observed a 2.5-fold increase in Mitfa staining intensity in *crestin:eGFP*-positive cells, indicating inhibition of BMP signaling leads to an increase in Mitfa protein in pigment progenitor cells at a single-cell level (*Figure 7C*). Furthermore, those cells that were Mitfa-positive and *crestin:eGFP*-negative showed a 1.7-fold increase in Mitfa staining intensity, indicating inhibition of BMP signaling also leads to an increase in Mitfa protein following specification of pigment cells (*Figure 7B and C*). Together, these results indicate BMP signaling suppresses *mitfa* expression in cells during specification of pigment cell lineages.

## Regulation of pigment cell fate by BMP signaling is dependent on *mitfa*

If deregulated *mitfa* expression is critical to the phenotypic defects observed upon inhibition of BMP signaling, then these defects should be dependent on *mitfa* function. To determine whether *mitfa* is indeed responsible for mediating the shift in cell fate regulated by BMP activity, we treated *mitfa(lf)* embryos with BMPi. As *mitfa* is necessary for the specification of all body melanocytes, *mitfa(lf)* animals do not develop any melanocytes during embryogenesis or through adulthood. However, these animals can develop iridophores and develop a greater number of iridophores at baseline than their wild-type counterparts (*Lister et al., 1999*). We hypothesized that, if an elevation of *mitfa* expression in BMPi-treated embryos was required to shift pigment progenitor cell fates from iridophores to melanocytes, there would be no decrease in the number of iridophores when *mitfa(lf)* embryos were treated with BMPi. Indeed, BMPi-treated embryos showed no difference in the number of iridophores compared to vehicle-treated controls (*Figure 7D and E*). Together, these results indicate that BMP inhibition requires *mitfa* to direct pigment progenitor cells away from iridophore fate.

## Discussion

Our results elucidate a role for *gdf6a*-activated BMP signaling in suppressing melanocyte development from the neural crest during embryogenesis. Inhibition of BMP signaling leads to an increase of neural crest cells expressing *mitfa*, affecting the proportion of neural crest cells specified as pigment progenitor cells. Additionally, in BMP-inhibited embryos these *mitfa*-positive pigment progenitor cells demonstrate an increased propensity to become melanocytes, instead of iridophores. Cells in BMP-inhibited embryos have increased expression of *mitfa*, and the function of *mitfa* is required for the reduction of iridophores observed in BMP-inhibited embryos. Based on these findings, we propose that *gdf6a*-activated BMP signaling normally represses *mitfa* expression, limiting both the development of pigment progenitor cells from the neural crest and the specification of melanocytes from these pigment progenitor cells. As discussed below, *MITF* is downregulated by *GDF6*-activated

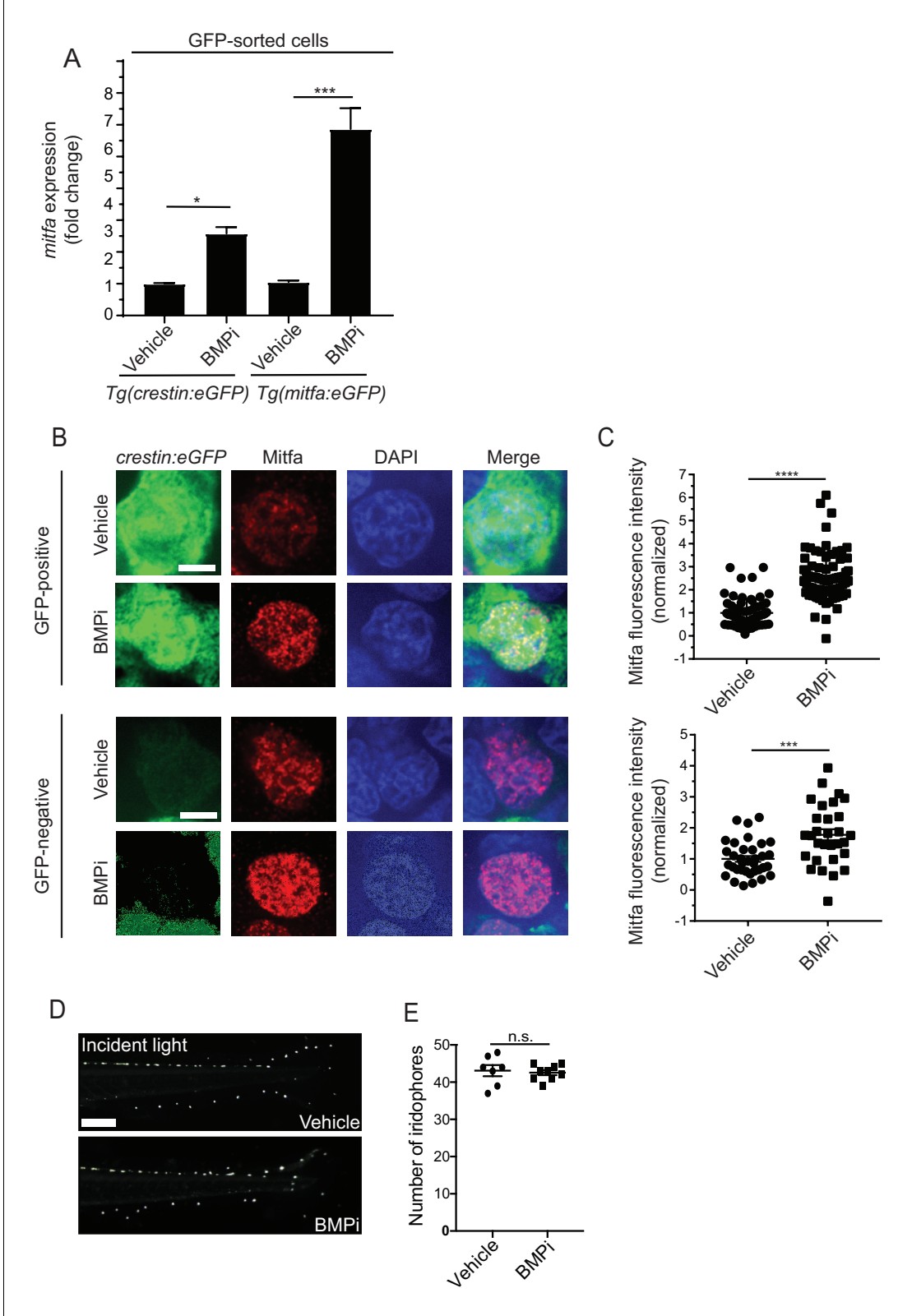

**Figure 7.** BMP signaling regulates expression of and acts through *mitfa* to impact pigment cell fates. (**A**) *mitfa* expression in sorted GFP-positive cells from *Tg(crestin:eGFP)* and *Tg(mitfa:eGFP)* embryos treated with vehicle or BMPi from 12 to 24 HPF, n = 4–5 replicates for each condition from two independent experiments (N = 2). (**B**) anti-Mitfa immunofluorescence, DAPI and merged images of *Tg(crestin:eGFP)* embryos treated with vehicle control or BMPi in GFP-positive cells (top) and GFP-negative cells (bottom), scale bar = 5 μm. (**C**) Quantification of anti-Mitfa fluorescence intensity of

*Figure 7 continued on next page*

*Figure 7 continued*

individual nuclei in GFP-positive cells (top) and GFP-negative cells (bottom); n = 65 and 74 for GFP-positive vehicle and BMPi groups, respectively; n = 35 and 30 for GFP-negative vehicle and BMPi groups, respectively, from three independent experiments (N = 3). (D) Incident light images of *mitfa (lf)* embryonic zebrafish treated with vehicle or BMPi from 12 to 24 HPF and imaged at 5 DPF, scale bar = 1 mm. (E) Quantification of dorsal iridophores in *mitfa(lf)* embryonic zebrafish treated with vehicle or BMPi from 12 to 24 HPF, n = 7 and 9 for vehicle and BMPi groups, respectively, from two independent experiments (N = 2). Error bars represent mean + /- SEM, P-value was calculated using one-way ANOVA with Tukey's multiple comparisons test in panel A and Student's t-test in panel C and E. *p<0.05, ***p<0.001, ****p<0.0001, n.s., not significant.

The online version of this article includes the following source data and figure supplement(s) for figure 7:

**Source data 1.** BMP signaling regulates expression of and acts through *mitfa* to impact pigment cell fates.
**Figure supplement 1.** Diagram of synteny between *MITF* and *mitfa* loci and SMAD binding motifs at the *mitfa* locus.

BMP signaling to prevent melanocytic differentiation in melanomas (*Venkatesan et al., 2018*). The function we have defined for *gdf6a*-activated BMP signaling in development suggests that its activity is co-opted in tumors to prevent differentiation of melanoma cells.

## Regulation of pigment cell fate by BMP signaling

Our studies indicate *gdf6a*-activated BMP signaling can regulate pigment cell development from the neural crest in two ways. First, BMP signaling restricts the number of neural crest cells that transition into *mitfa*-positive pigment cell progenitors. When BMP signaling is abrogated, additional cells adopt a pigment progenitor fate, which likely is a source of supernumerary melanocytes. Second, BMP signaling biases the fate choice of *mitfa*-positive progenitor cells. In BMP-deficient embryos, *mitfa*-positive progenitor cells more often become melanocytes and less often become iridophores. Previous studies have suggested a common melanocyte-iridophore progenitor (*Curran et al., 2010*; *Curran et al., 2009*; *Petratou et al., 2018*), and our data support the existence of such a progenitor and indicate that it is *mitfa*-expressing and influenced by BMP signaling. These studies implicate *foxd3* (*Curran et al., 2010*; *Curran et al., 2009*) or some other unknown factor (*Petratou et al., 2018*) that regulates the fate of this progenitor. Our data indicate BMP signaling can act independently of *foxd3* in impacting melanocyte development, thus suggesting BMP signaling may directly suppress the melanocyte fate or act indirectly through another factor. While BMP signaling regulates the fate of a common melanocyte-iridophore precursor, the decrease in the number of iridophores cannot fully account for the number of melanocytes gained in *gdf6a(lf)* and BMPi-treated embryos. Because *gdf6a(lf)* and BMPi-treatment are potentially impacting the entirety of neural crest development, other neural crest cells may be mis-specified to the melanocyte lineage. This mis-specification could account for the discrepancy between the gain of melanocytes and loss of iridophores. If mis-specification of other neural crest cells is occurring, other neural crest lineages could show a deficit. However, in our assays evaluating other lineages, we detected no deficits outside of a loss of iridophores. Among several possibilities, the deficit may be present in a neural crest lineage we did not directly measure. Alternatively, deficits in other neural crest lineages may be small and distributed across multiple other lineages, such that our assays are unable to detect those subtle changes. Lastly, proliferation within the neural crest and of neural crest derivatives following migration from the crest is known to occur (*Dougherty et al., 2013*; *Gianino et al., 2003*), and it is possible that such proliferation could compensate for any deficit. In summary, the supernumerary melanocytes observed in *gdf6a(lf)* and BMPi-treated embryos are likely to arise from some combination of neural crest cells that are shunted to the pigment cell lineage and melanocyte-iridophore precursors that preferentially adopt a melanocyte fate.

## Regulation of *mitfa* by BMP signaling

Our studies identify *gdf6a*-activated BMP signaling as a regulator of *mitfa* during pigment cell development in zebrafish. Previous studies have identified roles for *gdf6a* in the preplacodal ectoderm, retinal cell survival, and craniofacial development in zebrafish, while others have broadly connected BMP signaling to fate determination and cell survival in the neural crest in other model systems (*French et al., 2009*; *Gosse and Baier, 2009*; *Hanel and Hensey, 2006*; *Jin et al., 2001*; *Reed and Mortlock, 2010*; *Reichert et al., 2013*). However, the specific role of BMP signaling and of *gdf6a* on pigment cell development has heretofore been uncharacterized. Our analyses indicate that *gdf6a* is

expressed in neural crest cells prior to the anteroposterior onset of *mitfa* expression. In addition, we observed an overlap of BMP activity and *mitfa* expression at the leading edge of the anteroposterior *mitfa* progression. When BMP signaling was inhibited, we found increased expression in neural crest cells of *mitfa* RNA and Mitfa protein. Together, these results suggest that *gdf6a*-driven BMP signaling regulates expression of *mitfa* and, consequently, directs fates adopted by *mitfa*-expressing cells. We speculate that such a role underlies the excess melanocytes observed in *gdf6a(lf)* and BMPi-treated embryos. In the absence of *gdf6a* and BMP signaling, increased expression of *mitfa* could lead to a greater proportion of neural crest cells adopting a pigment cell fate and could lead to a greater propensity of melanocyte-iridophore precursors adopting a melanocyte fate. These findings are consistent with what has previously been established in human melanoma cells, where *GDF6*-activated BMP signaling has been shown to promote pSMAD binding to *MITF* and is suspected to directly regulate *MITF* expression (*Venkatesan et al., 2018*). Our results support this regulatory role and provide a developmental context in vivo to understand why *GDF6*-activated BMP signaling is able to regulate *MITF* in melanoma cells.

### Reiteration of normal physiologic function in melanoma

*GDF6* and BMP signaling were previously described in melanoma to suppress differentiation through binding of pSMAD to *MITF* and corresponding repression of *MITF* expression (*Venkatesan et al., 2018*). Results from the current study indicate *gdf6a* and BMP signaling likely act in a similar fashion during development to repress expression of *MITF*, either directly or indirectly, leading to suppression of melanocyte specification and differentiation from the neural crest. Together, these findings suggest BMP activity in melanoma is a recapitulation of normal regulatory functions executed by *gdf6a* and BMP signaling during pigment cell development. It has been previously established that lineage programs can be co-opted by cancers to promote pro-tumorigenic characteristics (*Carreira et al., 2006*; *Gupta et al., 2005*). These programs activate EMT factors, such as *TWIST1* and *SNAI2*, and factors associated with neural crest multipotency, such as *SOX10*, to promote invasiveness, proliferative capacity, metastatic capability, and therapeutic resistance (*Caramel et al., 2013*; *Casas et al., 2011*; *Shakhova et al., 2015*). However, it is unclear if these factors have similar regulation between normal development and melanoma. Here, we have described a developmental role for *GDF6* that is reiterated in a pathologic process in disease. Because initiation and maintenance of neural crest gene expression has been shown to be important in melanoma, a better understanding of how regulation occurs during development may have clinical implications (*Kaufman et al., 2016*). Our findings indicate BMP signaling has a regulatory role over key differentiation genes during melanocyte development from the neural crest. Many studies have implicated expression of neural crest and melanocyte factors during many phases of melanoma, including initiation, progression, invasion, metastasis, and therapeutic resistance of melanoma (*Carreira et al., 2006*; *Fallahi-Sichani et al., 2017*; *Gupta et al., 2005*; *Kaufman et al., 2016*; *Shaffer et al., 2017*). Taken together, these findings suggest therapeutic targeting of GDF6 or BMP signaling would likely have a positive impact on prognosis and outcome in melanoma patients by promoting differentiation in tumors.

## Materials and methods

**Key resources table**

| Reagent type (species) or resource | Designation | Source or reference | Identifiers | Additional information |
|---|---|---|---|---|
| Strain, strain background (*Danio rerio*) | *gdf6a(lf)* | *Gosse and Baier, 2009* PMID: 19164594 | | *gdf6a$^{s327}$* allele |
| Strain, strain background (*Danio rerio*) | *gdf6b(lf)* | This paper | | Generated using Golden Gate TALEN kit – see Materials and methods |

*Continued on next page*

*Continued*

| Reagent type (species) or resource | Designation | Source or reference | Identifiers | Additional information |
|---|---|---|---|---|
| Strain, strain background (*Danio rerio*) | *Tg(mitfa:eGFP)* | *Curran et al., 2009* PMID: 19527705 | | |
| Strain, strain background (*Danio rerio*) | *Tg(crestin:eGFP)* | *Kaufman et al., 2016* – PMID:26823433 | | |
| Strain, strain background (*Danio rerio*) | *Tg(crestin:mCherry)* | *Kaufman et al., 2016* PMID:26823433 | | |
| Strain, strain background (*Danio rerio*) | *Tg(ubi:switch)* | *Mosimann et al., 2011* PMID: 21138979 | | |
| Strain, strain background (*Danio rerio*) | *Tg(mitfa:gdf6a)* | This paper | | *gdf6a* expressed under the *mitfa* promoter, generated using Tol2Mediate transgenesis – see Materials and methods. |
| Strain, strain background (*Danio rerio*) | *mitfa(lf)* | *Lister et al., 1999* PMID: 10433906 | | *nacre* mutant |
| Strain, strain background (*Danio rerio*) | *foxd3(lf)* | *Stewart et al., 2006* PMID: 16499899 | | *sym1* mutant |
| Antibody | Phospho-SMAD1/5/9, rabbit monoclonal | CellSignaling | #13820 | 1:200 |
| Antibody | Mitfa, rabbit polyclonal | Venkatesan et al. | N/A | 1:100 |
| Antibody | GFP, mouse monoclonal | Thermo-Fisher | #MA5-15256 | 1:500 |
| Antibody | Goat anti-Mouse IgG-Alexafluor 488 | Thermo-Fisher | #A-11001 | 1:300 |
| Antibody | Goat anti-Rabbit IgG-Alexafluor 555 | Thermo-Fisher | #A-21428 | 1:300 |
| Antibody | Anti-DIG-AP Fab Fragments | Roche | #11093274910 | 1:1000 |
| Antibody | HuC/HuD antibody, mouse monoclonal | Thermo-Fisher | #A-21271 | 1:100 |
| Recombinant DNA reagent | pENTRP4P1r-mitfa | *Ceol et al., 2011* – PMID: 21430779 | | |
| Recombinant DNA reagent | pDONR221-gdf6b | *Venkatesan et al., 2018*– PMID: 29202482 | | |
| Recombinant DNA reagent | pDONR221-gdf6a | This paper | | Tol2 p221 entry vector containing *gdf6a* |
| Recombinant DNA reagent | pDONR221-Cre-ERT2 | *Mosimann et al., 2011* PMID: 21138979 | | |
| Recombinant DNA reagent | miniCoopR | *Ceol et al., 2011* PMID: 21430779 | | |
| Recombinant DNA reagent | pcsDest2 | *Villefranc et al., 2007* PMID: 17948311 | | |
| Recombinant DNA reagent | p3E-polyA | Tol2Kit (*Kwan et al., 2007*) PMID: 17937395 | | |
| Recombinant DNA reagent | pME-eGFP | Tol2Kit (*Kwan et al., 2007*) PMID: 17937395 | | |

*Continued on next page*

*Continued*

| Reagent type (species) or resource | Designation | Source or reference | Identifiers | Additional information |
|---|---|---|---|---|
| Recombinant DNA reagent | pDestTol2CG2 (395) | Tol2Kit (*Kwan et al., 2007*) PMID: 17937395 | | |
| Recombinant DNA reagent | pDestTol2pA2 (394) | Tol2Kit (*Kwan et al., 2007*) PMID: 17937395 | | |
| Recombinant DNA reagent | miniCoopR-mitfa: dnBMPR:pA | This paper | | miniCoopR vector expressing dominant negative BMP receptor; generated by multisite Gateway |
| Recombinant DNA reagent | miniCoopR-mitfa:eGFP:pA | This paper | | miniCoopR vector expressing control eGFP; generated by multisite Gateway |
| Recombinant DNA reagent | miniCoopR-mitfa: SMAD1-DVD:pA | This paper | | miniCoopR vector expressing constitutively active SMAD1; generated by multisite Gateway |
| Recombinant DNA reagent | 395-mitfa:gdf6a:pA | This paper | | pDEST vector expressing gdf6a under mitfa promoter; generated by multisite Gateway |
| Recombinant DNA reagent | 395-mitfa:Cre-ERT2:pA | This paper | | pDEST vector expression CreERT2 under mitfa promoter; generated by multisite Gateway |
| Recombinant DNA reagent | pcsDest2-gdf6a | This paper | | pDEST vector used to generate gdf6a probes; generated by Gateway reaction |
| Recombinant DNA reagent | pcsDest2-gdf6b | This paper | | pDEST vector used to generate gdf6b probes; generated by Gateway reaction |
| Sequence-based reagent | qPCR and Genotyping Primers | | | See *Supplementary file 1* for primer sequences |
| Commercial assay or kit | LR Clonase 2+ Kit | Thermo-Fisher | #12538120 | |
| Commercial assay or kit | LR Clonase Kit | Thermo-Fisher | #11791043 | |
| Commercial assay or kit | Golden Gate TALEN and TAL Effector Kit 2.0 | Addgene | #1000000024 | |
| Commercial assay or kit | mMessage mMachine Kit | Ambion | #AM1340 | |
| Commercial assay or kit | DIG RNA Labeling Kit | Roche | #11175025910 | |
| Commercial assay or kit | SuperScript III First Strand Synthesis | Thermo-Fisher | #18080051 | |
| Commercial assay or kit | SYBR Green Master Mix | Applied Biosystems | 4344463 | |
| Commercial assay or kit | Click-it EdU Cell Proliferation Kit for Imaging, AF 555 | Invitrogen | C10338 | |
| Chemical compound, drug | DMH1 | Sigma Aldrich | #D8946 | |

*Continued on next page*

Continued

| Reagent type (species) or resource | Designation | Source or reference | Identifiers | Additional information |
|---|---|---|---|---|
| Chemical compound, drug | Hydroxytamoxifen (4-OHT) | Sigma Aldrich | #H7904 | |
| Chemical compound, drug | Epinephrine | Acros Organics | #430140250 | |
| Chemical compound, drug | DAPI | Life Technologies | #D1306 | |
| Chemical compound, drug | Hoechst-33342 | Life Technologies | #H3570 | |
| Chemical compound, drug | Pronase | Sigma Aldrich | #10165921001 | |
| Chemical compound, drug | Trizol | Ambion | #15596026 | |
| Software, algorithm | Microsoft Excel | | | Microsoft Excel (Office 2016), www.microsoft.com |
| Software, algorithm | Graphpad Prism 7 | | | GraphPad Prism seven for Windows, GraphPad Software, www.graphpad.com |
| Software, algorithm | Image J | | | (*Schindelin et al., 2012*) |
| Software, algorithm | Leica LAS X | | | Leica LAS X for Windows, Leica Microsystems, www.leica-microsystems.com |
| Software, algorithm | FlowJo | | | Flow Jo v10 for Windows, Beckton, Dickinson and Company, www.flowjo.com |

## Zebrafish

Zebrafish were handled in accordance with protocols approved by the University of Massachusetts Medical School IACUC. Fish stocks were maintained in an animal facility at 28.5°C on a 14 hr/10 hr Light/Dark cycle (*Westerfield, 1995*). The wild-type strain used was AB. Published strains used in this study include *gdf6a(lf)* (*gdf6a*$^{s327}$) (*Gosse and Baier, 2009*), *Tg(mitfa:eGFP)* (*Curran et al., 2009*), *Tg(crestin:eGFP)* (*Kaufman et al., 2016*), *Tg(crestin:mCherry)* (*Kaufman et al., 2016*), *mitfa (lf)* (*Lister et al., 1999*), *Tg(ubi:switch)* (*Mosimann et al., 2011*), *Tg(aox5:PALM-eGFP)* (*Eom and Parichy, 2017*). Construction of new strains generated are detailed below.

## DNA constructs

DNA constructs were built using Gateway cloning (Life Technologies). Sequences of *gdf6a*, *dnBMPR* (*Pyati et al., 2005*) and *SMAD1-DVD* (*Nojima et al., 2010*) were PCR-amplified and cloned into pDONR221 (Life Technologies). Oligonucleotides used in cloning are described in Key Resources Section. Previously published entry clones used in this study were pENTRP4P1r-*mitfa* (*Ceol et al., 2011*), pDONR221-*gdf6b* (*Venkatesan et al., 2018*), pDONR221-*CreERT2* (*Mosimann et al., 2011*). Previously published destination vectors used in this study are MiniCoopR (MCR) (*Ceol et al., 2011*) and pcsDest2 (*Villefranc et al., 2007*). p3E-*polyA*, pME-*eGFP*, pDestTol2CG2, pDestTol2pA2, pCS2FA-transpoase were acquired from the Tol2Kit (*Kwan et al., 2007*). Using the entry clones and destination vectors described above, the following constructions were built using multisite or single site Gateway (Life Technologies): MCR-mitfa:dnBMPR:pA, MCR-mitfa:eGFP:pA, MCR-mitfa:SMAD1-DVD:pA, pDestTol2CG2-mitfa:gdf6a:pA, pDestTol2pA2-mitfa:CreERT2:pA, pcsDest2-gdf6a, pcsDest2-gdf6b. All constructs were verified by restriction digest or sequencing.

## Construction of *gdf6b(lf)*

To generate *gdf6b(lf)* mutants, we used TALEN genome editing. TALEN's were designed targeting exon 1 of *gdf6b* (TAL1 sequence: GTCAGCATCACTGTTAT; TAL2 sequence: CCTTGATCGCCCTTC T). TALENs were assembled using the Golden Gate TALEN kit (Addgene) per the manufacturer's instructions. TALEN plasmids were linearized and transcribed with mMESSAGE mMACHINE kit (Ambion). Zebrafish embryos were injected with 50 pg of mRNA of each TALEN arm. Injected embryos (F0) were matured to breeding age and outcrossed. Resulting offspring (F1) were geno-typed by extraction genomic DNA from fin clips per standard protocol and PCR amplification with *gdf6b* primers. F1 offspring carrying mutations by genotyping were sequenced to identify mutations predicted to lead to loss of function of *gdf6b*. Following identification of candidate zebrafish by sequencing, zebrafish were bred to generate homozygous *gdf6b(lf)* mutations. Whole RNA was isolated from homozygous *gdf6b(lf)* embryos at 20 HPF and qRT-PCR was used to determine effective depletion of *gdf6b* transcripts. Primers for genotyping and qRT-PCR are listed in the Key Reagents section.

## Construction of *Tg(mitfa:gdf6a)*

To generate the *Tg(mitfa:gdf6a)* transgenic line, 25 pg of pDestTol2CG2-mitfa:gdf6a:pA was injected along with 25 pg of Tol2 transposase RNA, synthesized from pCS2FA-*transposase*, into single cell wild-type embryos (*Kwan et al., 2007*). Embryos were screened for incorporation of the transgene by expression of cmlc:eGFP in the heart at 48 HPF. Animals with eGFP-positive hearts (F0) were outcrossed to wild-type animals to determine germline incorporation.

## Drug treatments

Drugs used in experiments were reconstituted at stock concentrations in solvent as follows: DMH1 (BMPi), 10 mM in DMSO; Tamoxifen (4-OHT), 1 mg/mL in ethanol; Epinephrine, 10 mg/mL in embryo media. Embryos were dechorionated by incubating in Pronase (Roche) for 10 min with gentle shaking. Dechorionated embryos were transferred to 6-well plates coated in 1.5% agarose in embryo media. Embryo media with appropriate drug concentration or vehicle control was added to each well. For BMPi and 4-OHT treatments, embryos were treated from 12 HPF (6ss) to 24 HPF (Prim-5). Embryos were incubated at 28.5°C for the duration of the drug treatment. Following drug treatment, embryos were thoroughly washed in fresh embryo medium and returned to incubator in new embryo medium until analysis.

## Lineage tracing

To trace the lineage of embryonic pigment cells, *Tg(ubi:switch)* embryos were injected with 25 pg of pDestTol2pA2-mitfa:Cre-ERT2:pA and 25 pg of Tol2 transposase RNA at the single-cell stage. At 12 HPF, injected embryos were treated with BMPi and 4-OHT as described above. Following treatment, embryos were thoroughly washed and allowed to mature at 28.5°C to 5 DPF. Embryos were treated with 1 mg/mL epinephrine to contract melanosomes, anesthetized using 0.17 mg/mL tricaine in embryo media, mounted in 1% low-melt agarose on a plastic dish, and submerged in embryo media for imaging.

## Mosaic rescue

MiniCoopR constructs MCR-mitfa:dnBMPR:pA, MCR-mitfa:SMAD1-DVD:pA, and MCR-mitfa:eGFP: pA (control) were used. *mitfa(lf)* animals were injected with 25 pg of a single construct and 25 pg of Tol2 transposase RNA. Upon successful integration of the MCR constructs, the *mitfa*-minigene in the construct allowed development of melanocytes. Embryos were screened for incorporation of the transgene by rescue of melanocytes at 5 DPF (*Ceol et al., 2011*).

## In Situ Hybridization

RNA sense and anti-sense probes were synthesized from pcsDest2-*gdf6a* and pcsDest2-*gdf6b* constructs using DIG RNA Labeling Kit (Roche) per the manufacturer's instruction. Wild-type embryos of the appropriate stage were fixed in 4% PFA at 4°C for 24 hr. Following fixation, embryos were dehydrated in methanol at stored at −20°C. Whole mount in situ hybridization was performed as previously described (*Reichert et al., 2013*). Hybridized probes were detected using anti-digoxigenin

(DIG) antibodies tagged with alkaline-phosphatase (AP) (Roche) using NBT/BCIP (Roche) solution per the manufacturer's instructions. Stained embryos were mounted in 2.5% methylcellulose and imaged using a Leica M165FC microscope and Leica DFC400 camera. Specificity of the probes was verified using sense probes synthesized from the same construct.

### EdU incorporation

Embryos were dechorionated at the desired time and thoroughly washed in embryo media. Embryos were transferred to 1 mM EdU (Invitrogen), 10% DMSO in embryo media and incubated on ice for 1 hr, then incubated at 28.5°C for 1 hr until the desired stage was reached. Embryos were washed thoroughly in fresh embryo media and fixed in 4% PFA for 2 hr. Following fixation, embryos were permeabilized by washing with 1% DMSO, 1% Triton X-100 in PBS for 1 hr. The EdU reaction mix was prepared per the manufacturer's instructions. Embryos were then transferred to the reaction mix and incubated in the dark for 1 hr at room temperature. Following the reaction, embryos were washed in PBST and mounted for imaging. Cells were counted and data were analyzed using Microsoft Excel and GraphPad Prism 7.

### Immunofluorescence

Embryos were fixed at the desired time or following drug treatment in 4% PFA for 24 hr at 4°C. Whole mount immunofluorescence was performed as previously described (*Venkatesan et al., 2018*). Primary antibodies used were pSMAD-1/5/8 (1:100 dilution) (Cell Signal Technologies), HuC/D (1:100 dilution) (Sigma), mitfa (1:100 dilution) (*Venkatesan et al., 2018*). AlexaFluor-488 (Invitrogen) and AlexaFluor-555 (Invitrogen) conjugated secondary antibodies were used to detect primary antibody signaling. Nuclei were counterstained with DAPI. Following staining, animals were dissected to remove yolk sack and flat mounted laterally on slides using VectaShield mounting medium. Fluorescent images were taken using a Leica DM5500 microscope with a Leica DFC365FX camera, and a Zeiss Axiovert 200 microscope outfitted with a Yokogawa spinning disk confocal scanner. Cells and structures were counted, and data was analyzed using Microsoft Excel and GraphPad Prism 7.

### Flow cytometry and Fluorescence Activated Cell Sorting (FACS)

Embryos were treated and matured to appropriate age as per drug treatment protocol described above. At a desired timepoint, embryos were washed in PBS and transferred to 500 µL of PBS + 5% FBS (FACS buffer). Embryos were mechanically dissociated in FACS buffer using a mortar and pestle. Dissociated embryos were washed with FACS buffer and filtered through a 40 µm mesh membrane. Samples were analyzed using a BD FACS Aria II flow cytometer and sorted directly into Trizol LS (Life Technologies) for RNA isolation. Flow cytometry data was analyzed using FlowJo software (Becton, Dickinson and Company) and GraphPad Prism 7.

### Quantitative Real-Time PCR (qRT-PCR)

Oligos used for qRT-PCR primers are listed in Key Reagents section. RNA was isolated from FACS-sorted cells or whole embryos using Trizol reagent (Life Technologies) and purified using the RNeasy kit (Quiagen) per manufacturer's protocol. cDNA was synthesized from purified RNA using the SuperScript III First Strand Synthesis kit (Thermo Fisher). Reaction mixes were assembled with SYBR Green RT-PCR master mix (Thermo Fisher), primers, and 25 ng cDNA, and analyzed using a StepOnePlus Real Time PCR System (Applied Biosystems). All samples were normalized to *β-actin*, unless otherwise noted, and fold changes were calculated using the ΔΔCt method using Microsoft Excel and GraphPad Prism 7.

### Imaging and quantification

Zebrafish adults and embryos were treated with 1 mg/mL epinephrine to contract melanosomes prior to imaging unless otherwise noted. Fish were anesthetized in 0.17% Tricaine in embryo media and positioned in 2.5% methylcellulose in embryo media for imaging. Images of adult fish were captured with a Nikon D90 DSLR camera. Brightfield and incident light images of embryos were captured with Leica M165FC microscope and Leica DFC400 camera. Fluorescent images of embryos were captured with a Leica DM5500 upright microscope with a Leica DFC365FX camera, and a Zeiss Axiovert 200 microscope outfitted with a Yokogawa spinning disk confocal scanner. Images were

processed using ImageJ and Leica LAS X software. Cells were counted and analyses were performed using Microsoft Excel and GraphPad Prism 7. Statistical calculations were performed using GraphPad Prism seven as described in each Figure legend.

## Statistical analysis

Statistical analyses were performing using GraphPad Prism seven software package. Statistical significance of experiments was calculated using Student's t-test, ratio-paired t-test, Fisher's exact test with Bonferroni's correction, 1-way ANOVA with Tukey's multiple comparison test as described in each figure legend. Statistical significance was denoted as follows: not significant (ns) $p > 0.05$, $*p < 0.05$, $**p < 0.01$, $***p < 0.001$ and $****p < 0.0001$.

# Acknowledgements

We thank Nathan Lawson for the pcsDest2 plasmid; David Kimelman for the dnBMPR plasmid; Takenobu Katagiri for the SMAD1-DVD plasmid; Christian Mossiman for the *Tg(ubi:switch)* zebrafish strain and CreERT2 plasmid; Charles Kaufmann for *Tg(crestin:eGFP)* and *Tg(crestin:mCherry)* zebrafish strains; Thomas Look for the *foxd3(zdf10)* zebrafish strain; Patrick White, Ed Jaskolski and the staff at the UMMS Animal Medicine Department for fish care; Tammy Krumpoch for guidance and assistance in performing flow cytometry and FACS experiments. AKG was supported by a Melanoma Research Foundation Looney Legacy Foundation Medical Student Award, Center for Translational Sciences TL1 Training Fellowship through the UMMS CCTS (UL1-TR001453), NCI NRSA F31 Predoctoral Fellowship (1F31CA239478-01). Research was supported by a Kimmel Scholar Award (SKF-13–123), Department of Defense Peer Reviewed Cancer Research Program Career Development Award (W8IXWH-13–0107) and NIH National Institute of Arthritis and Musculoskeletal and Skin Diseases grant (R01AR063850) to CJC. The content is solely the responsibility of the authors and does not necessarily represent the official views of the Department of Defense or NIH.

# Additional information

### Funding

| Funder | Grant reference number | Author |
| --- | --- | --- |
| Melanoma Research Foundation | | Alec K Gramann |
| National Cancer Institute | 1F31CA239478-01 | Alec K Gramann |
| National Center for Advancing Translational Sciences | UL1-TR001453 | Alec K Gramann |
| Congressionally Directed Medical Research Programs | W8IXWH-13-0107 | Craig J Ceol |
| National Institute of Arthritis and Musculoskeletal and Skin Diseases | R01AR063850 | Craig J Ceol |
| Sidney Kimmel Foundation for Cancer Research | SKF-13-123 | Craig J Ceol |

The funders had no role in study design, data collection and interpretation, or the decision to submit the work for publication.

### Author contributions

Alec K Gramann, Conceptualization, Data curation, Formal analysis, Funding acquisition, Investigation, Methodology, Writing - original draft, Writing - review and editing, AKG helped conceive the project; design and interpret melanocyte quantification experiments and results; design and interpret the proliferation, specification, and neural crest lineage experiments and results; perform the flow cytometry, qRT-PCR, immunofluorescence, lineage tracing, and in situ hybridization experiments; perform the melanocyte quantification experiments; and write the manuscript; Arvind M Venkatesan, Conceptualization, Data curation, Formal analysis, Investigation, Methodology, Writing -

review and editing, AMV helped conceive the project; design and interpret the melanocyte quantification experiments and results; generate the MiniCoopR and pCS-DEST plasmids; generate probes for in situ hybridization; and generate and isolate the gdf6b mutant zebrafish; Melissa Guerin, Investigation, Methodology, MG helped generate and isolate the gdf6b mutant zebrafish; Craig J Ceol, Conceptualization, Formal analysis, Supervision, Funding acquisition, Investigation, Methodology, Writing - original draft, Project administration, Writing - review and editing, CJC helped conceive the project; design and interpret the melanocyte quantification experiments and results; design and interpret the proliferation, specification, and neural crest lineage experiments and results; and write the manuscript

### Author ORCIDs
Alec K Gramann (iD) https://orcid.org/0000-0001-7527-5533
Craig J Ceol (iD) https://orcid.org/0000-0002-7188-7580

### Ethics
Animal experimentation: This study was performed in strict accordance with the recommendations in the Guide for the Care and Use of Laboratory Animals of the National Institutes of Health. Zebrafish were handled in accordance with protocols approved by the University of Massachusetts Medical School IACUC protocol (A-2171-19). For procedures, including imaging and genotyping, animals were anesthetized in 0.17% tricaine or euthanized by overdose of tricaine. Every effort was made to minimize suffering.

### Decision letter and Author response
Decision letter https://doi.org/10.7554/eLife.50047.sa1
Author response https://doi.org/10.7554/eLife.50047.sa2

## Additional files

### Supplementary files
• Supplementary file 1. Primers used for qRT-PCR and genotyping reactions. (A) Primers used for qRT-PCR reactions. (B) Primers used for genotyping PCR reactions and generation of Gateway compatible DNA fragments.

• Transparent reporting form

### Data availability
All data generated or analyzed during this study are included in the manuscript and supporting files. Source data files have been provided for all figures and supplements.

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
