## [Decision Letter]

**Acceptance summary:**

This manuscript describes the role of *gdf6a*-induced BMP signaling in regulating melanoblast specification during zebrafish development. This study describes two mechanisms by which BMP signaling acts to limit the number of neural crest cells that adopt a melanocyte fate, in particular how BMP signaling influences development of melanoblasts at the expense of iridophores by down-regulating expression of the key transcription factor *mitfa* that specifies melanocyte fate. BMP signaling in normal melanocyte development therefore mirrors its role in melanoma where it represses *MITF* expression leading to a less differentiated state and more aggressive melanoma. This study will be of general interest to developmental biology field as a novel mechanism regulating the fate of neural crest cells in zebrafish. Understanding the mechanism by which the phosphorylated SMAD transcription factor acts as a repressor of *mitfa* expression and hence a major determinant of neural crest cell fate remains a major outstanding question raised by this study.

**Decision letter after peer review:**

Thank you for submitting your article "Regulation of melanocyte development by ligand-dependent BMP signaling underlies oncogenic BMP signaling in melanoma" for consideration by *eLife*. Your article has been reviewed by three peer reviewers, including Irwin Davidson as the Reviewing Editor and Reviewer #1, and the evaluation has been overseen by Richard White as the Senior Editor. The reviewers have discussed the reviews with one another and the Reviewing Editor has drafted this decision to help you prepare a revised submission.

This manuscript describes the role of *gdf6a*-induced BMP signaling in regulating melanoblast specification during zebrafish development. By using genetic mutants, small molecule inhibitors, lineage tracing, and melanocyte differentiation rescue using the miniCoopR system, the authors describe two mechanisms by which BMP signaling acts to limit the number of neural crest cells that adopt a melanocyte fate, in particular how BMP signaling influences development of melanoblasts at the expense of iridophores by down-regulating *mitfa* expression. BMP signaling in normal melanocyte development therefore mirrors its role in melanoma where it represses *MITF* expression leading to a less differentiated state and more aggressive melanoma.

While all three referees found this study to be interesting and well done, several issues require further attention.

Essential revisions:

1) Figure 3A: In situs for *gdf6a* in the neural crest, and the absence from *mitfa* expression are not convincing. Figure 3B: It is not clear what the phopho-SMAD signal is being counted in the bottom embryo. What regions of the animal are defined as leading edge and rostral?

2) The authors present an adult phenotype for the *gdf6a* mutant and clearly state that "zebrafish develop their adult pigment pattern during metamorphosis, it is possible *gdf6a* acts during this stage to change adult pigmentation", although nothing is mentioned about the neural crest-derived cells established during somitogenesis at the level of the DRG and give rise to the adult melanocytes (Singh et al., 2015; Dooley et al., 2013; Hultman et al., 2009). The authors should explore this further.

3) In order to determine the proliferation activity of neural crest cells and pigment progenitor cells the authors measured the DNA content by fluorescence activated cell sorting. Although this gives an indication of the cell cycle phase in which cells are found at a certain moment, it doesn't tell us whether cells have higher proliferation rate which could be assessed using thymidine analogue incorporation. This would provide additional data that would support their conclusions, given the modest effects seen in Figure 2C and 2E.

4) One interesting aspect of this study and the previous study of *GDF6* regulation in melanoma is the repression of *MITF* expression. Here, repression of *mitfa* expression by BMP signaling is proposed to play a key role in the choice of melanocyte versus iridophore fate. The authors should better address the mechanism of *MITF* repression. They previously showed binding of phospho-SMAD to a region upstream of the *MITF-M* promoter in human melanoma cells. Is this element conserved in zebrafish? How does phospho-SMAD act as transcriptional repressor under these conditions? It has been shown that *foxd3*, like BMP signaling regulates the ratio of melanophores to iridophores. Curran et al., 2009, argue that *foxd3* acts upstream of *mitfa* expression. In this study, and in the Venkatesan et al., 2018 paper, BMP signaling stimulated phospho-SMAD acts upstream of *mitfa* expression. Do they act in parallel or in sequence? Is there a genetic interaction study that would help establish the epistasis? For instance, does *gdf6a(lf)* still increase melanocytes in *foxd3(lf)* mutants? These experiments would better help to distinguish between a direct repression by SMAD binding to a cis-regulatory element of *MITF* or an indirect mechanism.

5) The study contains little description of how BMP signaling affects the expression of key melanocyte genes other than *mitfa* and key neural crest genes aside those shown in Figure 5A. The authors should investigate for example whether *MITF* target genes are down-regulated along with *mitfa* and if and how *sox10* and *foxd3* expression is affected.

---

## [Author Response]

Essential revisions:1) Figure 3A: In situs for gdf6a in the neural crest, and the absence from mitfa expression are not convincing. Figure 3B: It is not clear what the phopho-SMAD signal is being counted in the bottom embryo. What regions of the animal are defined as leading edge and rostral?

We recognize that our initial description of *gdf6a* and *mitfa* in situ expression data should have been more thorough and precise. In the revised manuscript, we cover the domains and sites of expression for *gdf6a* and *mitfa* and note that these are consistent with previously described expression data (Lister et al., 1999; Reichert et al., 2013; Rissi et al., 1995). Neural crest expression of *gdf6a* is not detected by in situs at later embryonic time points. Since expression patterns for *gdf6a* and *mitfa* have not previously been juxtaposed, we simply use the in situ data to show that the domains of expression are ‘mostly, if not completely, non-overlapping’.

In addition to the clarification above, we have specifically examined *gdf6a* expression in neural crest cells. We isolated eGFP-positive neural crest cells by FACS from *Tg(crestin:eGFP)* embryos and performed qRT-PCR to measure *gdf6a* and *mitfa* expression (new Figure 3A). In this experiment, neural crest cells from early (14ss, 16hpf) embryos and late (26ss, 22hpf) embryos were isolated. We confirmed *gdf6a* expression in the neural crest and found that it was relatively higher at 16hpf as compared to the 22 hpf neural crest. By contrast, *mitfa* expression was relatively higher at 22hpf compared to 16hpf. These data complement the in situ studies, showing early *gdf6a* neural crest expression as well as an inverse correlation between *gdf6a* and *mitfa* expression, at least in this cell type.

To clarify pSMAD positivity, we have redrawn the zebrafish diagram to show rostral regions and the leading edge on a single embryo. The leading edge is categorized as the 5 distal-most *mitfa:GFP* positive cells. Rostral *mitfa:GFP* cells include other *mitfa:GFP* cells not in the 5 distal-most GFP positive cells. We have included this description in both the figure legend and within the body text.

2) The authors present an adult phenotype for the gdf6a mutant and clearly state that "zebrafish develop their adult pigment pattern during metamorphosis, it is possible gdf6a acts during this stage to change adult pigmentation", although nothing is mentioned about the neural crest-derived cells established during somitogenesis at the level of the DRG and give rise to the adult melanocytes (Singh et al., 2015; Dooley et al., 2013; Hultman et al., 2009). The authors should explore this further.

To address the reviewer comment, we have performed Hu C/D antibody staining in *Tg(mitfa:eGFP*) embryos at 3 DPF treated with BMPi or vehicle control (new Figure 5—figure supplement 1F). We observed an increase in the number of DRG-associated *mitfa:EGFP*-positive cells in BMPi-treated embryos. This suggests BMP inhibition not only increases embryonic melanocytes, but also impacts neural crest-derived melanocyte stem cells and their role in adult pigment pattern formation. This raises several interesting questions (e.g. what is the source of these extra cells – altered specification, proliferation, migration?; what are the fates that these cells adopt?) that are to be the subject of extensive further studies.

3) In order to determine the proliferation activity of neural crest cells and pigment progenitor cells the authors measured the DNA content by fluorescence activated cell sorting. Although this gives an indication of the cell cycle phase in which cells are found at a certain moment, it doesn't tell us whether cells have higher proliferation rate which could be assessed using thymidine analogue incorporation. This would provide additional data that would support their conclusions, given the modest effects seen in Figure 2C and 2E.

We have performed EdU incorporation during neural crest development of *Tg(crestin:eGFP)* and *Tg(mitfa:eGFP)* embryos to determine if any changes in proliferation rate are evident. During each pulse of EdU (12-14hpf, 16-18hpf, 20-22hpf) we observed similar proportions of EdU and eGFP double-positive cells in BMPi- and vehicle-treated groups, indicating no significant change in proliferation rate upon BMP inhibition. These results are included in Figure 2—figure supplement 1 and discussed in the Results section of the main text.

4) One interesting aspect of this study and the previous study of GDF6 regulation in melanoma is the repression of MITF expression. Here, repression of mitfa expression by BMP signaling is proposed to play a key role in the choice of melanocyte versus iridophore fate. The authors should better address the mechanism of MITF repression. They previously showed binding of phospho-SMAD to a region upstream of the MITF-M promoter in human melanoma cells. Is this element conserved in zebrafish? How does phospho-SMAD act as transcriptional repressor under these conditions? It has been shown that foxd3, like BMP signaling regulates the ratio of melanophores to iridophores. Curran et al., 2009, argue that foxd3 acts upstream of mitfa expression. In this study, and in the Venkatesan et al., 2018 paper, BMP signaling stimulated phospho-SMAD acts upstream of mitfa expression. Do they act in parallel or in sequence? Is there a genetic interaction study that would help establish the epistasis? For instance, does gdf6(lf) still increase melanocytes in foxd3(lf) mutants? These experiments would better help to distinguish between a direct repression by SMAD binding to a cis-regulatory element of MITF or an indirect mechanism.

In our previous work, we used ChIP-seq to identify a pSMAD binding region at the *MITF* locus, within the intron of long *MITF* isoforms, and upstream of *MITF-M* (Venkatesan et al., 2018). Within this binding region we found a known pSMAD binding motif (GC-SBM) (Morikawa et al., 2011). To assess whether there is a potential for pSMAD regulation of the *mitfa* locus in zebrafish, we first investigated whether the *MITF* and *mitfa* loci reside in syntenic regions of the genome (new Figure 7—figure supplement 1A). We found multiple orthologous genes flanking *MITF* and *mitfa*, indicating synteny at this region of the genome. We next examined whether the pSMAD binding region identified in human cells was conserved in zebrafish. Using BLAST and BLAT searches as well as pairwise DNA matrices between the pSMAD binding region and *mitfa* locus, we did not identify extensive tracts of DNA sequence conservation. However, within the *mitfa* locus in zebrafish, we identified multiple intronic and exonic pSMAD binding motifs, as well as multiple motifs within the *mitfa* promoter (new Figure 7—figure supplement 1B). These include the GC-SBM motif described above as well as additional SBMs described previously (Jonk et al., 1998). Given the synteny between the *MITF* and *mitfa* loci, the presence of SMAD binding motifs, and similar transcriptional changes under BMP inhibition, we believe it is possible *mitfa* is directly regulated by BMP-activated pSMAD.

To address the potential interaction of *foxd3* and *gdf6a* signaling in regulating the development of melanocytes, we first performed qPCR for *foxd3* expression in both *gdf6a*-mutant and BMPi-treated animals. Compared to controls, neither experimental group showed significant changes in *foxd3* expression (Figure 5—figure supplement 1). To assess if there is functional relationship between BMP signaling and *foxd3* in melanocyte development, we treated *foxd3(zdf10)*-mutant zebrafish with BMPi or vehicle, and assessed development of the melanocyte lineage. As previously described (Stewart et al., 2006), *foxd3(zdf10*) mutants have deficiencies in specific neural crest derived populations, but maintain a normal number of melanocytes during early development. In our experiment, we observed a comparable increase in the number of melanocytes between mutant, heterozygous, and wild-type *foxd3* embryos treated with BMPi. Since inhibition of BMP signaling (more melanocytes) is different than loss of *foxd3* (melanocytes unchanged), this indicates that BMP signaling is not acting exclusively with *foxd3* to regulate melanocyte fate. Consequently, the simplest interpretation is that BMP signaling can act independently of *foxd3* to regulate melanocyte development. We have included these results in Figure 5—figure supplement 1.

5) The study contains little description of how BMP signaling affects the expression of key melanocyte genes other than mitfa and key neural crest genes aside those shown in Figure 5A. The authors should investigate for example whether MITF target genes are down-regulated along with mitfa and if and how sox10 and foxd3 expression is affected.

We have performed qPCR for additional genes associated with melanocyte differentiation. We observed an upregulation of *mitfa* targets *tyr* and *mc1r* when *gdf6a* is lost or BMP signaling inhibited (now included in Figure 1E), supporting our previous findings for *mitfa* and *tyrp1b*. We have also performed qPCR for both *foxd3* and *sox10* (Figure 5—figure supplement 1). As discussed above, *foxd3* expression was unaltered in *gdf6a*-mutant or BMPi-treated embryos, and functional evaluation suggested BMP signaling can act independently of *foxd3* in melanocyte development. We observed a slight downregulation of *sox10* when *gdf6a* is lost or BMP signaling is inhibited, which is in line with our previous findings in melanoma (Venkatesan et al., 2018). *sox10* normally promotes *mitfa* expression. Since we still observed an increase of *mitfa* expression in *gdf6a(lf)* and BMPi-treated embryos, we speculate that the reduced *sox10* expression was not enough to impede its activation of *mitfa* or reduced *sox10* had some negative impact on *mitfa* expression but this impact was counteracted by loss of BMP-mediated *mitfa* repression.